# Valproic acid use is associated with diminished risk of contracting COVID-19, and diminished disease severity: Epidemiologic and *in vitro* analysis reveal mechanistic insights

**Amanda Watson**[1][☉], **Pankil Shah**[2][☉], **Doug Lee**[2], **Sitai Liang**[2], **Geeta Joshi**[2], **Ediri Metitiri**[2], **Wasim H. Chowdhury**[2], **Dean Bacich**[2], **Peter Dube**[3], **Yan Xiang**[4], **Daniel Hanley**[5], **Luis Martinez-Sobrido**[6], **Ronald Rodriguez**[7]*

1 Glenn Biggs Institute for Alzheimer's & Neurodegenerative Diseases, University of Texas Health Science Center San Antonio, San Antonio, Texas, United States of America, 2 Department of Urology, University of Texas Health Science Center San Antonio, San Antonio, Texas, United States of America, 3 Boehringer Ingelheim in Ames, Ames, Iowa, United States of America, 4 Department of Microbiology, Immunology and Molecular Genetics, University of Texas Health Science Center San Antonio San Antonio, Texas, United States of America, 5 Department of Neurology & Neurosurgery, Johns Hopkins School of Medicine, Baltimore, Maryland, United States of America, 6 Texas Biomedical Research Institute, San Antonio, Texas, United States of America, 7 Department of Medical Education, and Department of Urology, University of Texas Health Science Center San Antonio, San Antonio, Texas, United States of America

☉ These authors contributed equally to this work.

* Rodriguezr32@uthscsa.edu

**Data Availability Statement:** The electronic medical records data supporting this study's findings are available from OptumInsight Life

## Abstract

The SARS-CoV-2 pandemic has caused unprecedented worldwide infections from persistent mutant variants with various degrees of infectivity and virulence. The elusiveness of a highly penetrant, worldwide vaccination strategy suggests that the complete eradication of SARS-CoV-2 is unlikely. Even with the advent of new antiviral agents, the disease burden worldwide continues to exceed current preventative and therapeutic strategies. Greater interest has been placed towards the development of affordable,broadly effective antiviral therapeutics. Here, we report that the small branched-chain fatty acid Valproic acid (VPA), approved for maintenance of seizure and bipolar disorder, has a novel anti- coronavirus activity that can be augmented with the addition of a long-chain, polyunsaturated omega-3 fatty acid, Docosahexaenoic acid (DHA). An EMR-based epidemiological study of patients tested for COVID-19 demonstrated a correlation exists between a reduced infection rate in patients treated withVPA of up to 25%, as well as a decreased risk of emergency room visits, hospitalization, ICU admission,and use of mechanical ventilation. *In vitro* studies have demonstrated that VPA modifies gene expression in MRC5 cells. Interestingly, VPA correlates with the inhibition of several SARS-CoV2 interacting genes and the greater inhibition of alpha-coronavirus HCoV-229E (a "common cold" virus) and SARS-CoV2. The VPA-DHA combination activates pre-existing intracellular antiviral mechanisms normally repressed by coronaviruses. Gene expression profiles demonstrate subtle differences in overall gene expression between VPA-treated and VPA-DHA-treated cells. HCoV-229E infection caused

Sciences, Inc. ("Optum"). The data was obtained by the University of Texas (UT) Center for Health Care Data (CHCD) under a Data License Agreement for Research, used under an Interagency contract by the study authors, and is subject to restrictions. UT CHCD may be reached at CHCD.QA@uth.tmc. edu, and Optum can be reached at https://www. optum.com/en/business/life-sciences.html.

**Funding:** This work was funded by the National Center for Advancing Translational Sciences, TR004529 granted to Pankil Shah. The Abraham and Linda Littenberg Foundation, the DHR Endowment and the UT Health San Antonio Dean's Fund granted to Ronald Rodriguez. The funders had no role in study design, data collection and analysis, decision to publish, or preparation of the manuscript.

**Competing interests:** The authors have declared that no competing interests exist.

an intensely different response with a marked induction of multiple intracellular inflammatory genes. Changes in gene expression took at least 24 hours to manifest and most likely why prior drug screens failed to identify any antiviral VPA activity despite *in silico* predictions. This report demonstrates an interaction between HDAC inhibition and the potent activation of cellular antiviral responses. A foundation now exists for a low-cost, highly effective antiviral strategy when supplemented with DHA.

## Introduction

Valproic acid is a short branched-chain fatty acid, used initially as an excipient for many decades, as it was thought to be inert until it was discovered to have antiseizure activity in the early 1960s [1]. Subsequently, it was developed as a therapeutic for seizure disorders, and later bipolardisorder [2, 3], as well as migraine headaches. While it has a well-established toxicity profile, it has been found to elevate the risk of certain birth defects roughly three-fold and therefore, Valproic acid is contraindicated for use in pregnant women [4]. The mechanism of action in causing birth defects is not fully understood; however, the inhibition of HDAC (Histone deacetylases) activity has been proposed [5].

This HDAC inhibitory (HDACi) activity has been studied for potential use as a cancer therapeutics [6, 7] and as a treatment strategy for HIV, where it can stimulate the emergence of latent infection from a lysogenic state- theoretically permitting the potential for eradication of disease when used in combination with HIV Highly Active Anti-retroviral Therapy (HAART) [8]. Valproic acid has been found to have antiviral activity for various viruses, to which DNA, RNA,and enveloped viruses appear to have the most susceptibility [9].

In order to identify coronavirus antiviral therapeutics, many FDA-approved drugs were assessed for potential repurposing against COVID-19. Initial results demonstrated dozens of potential agents predicted to impact coronavirus replication [10–13]. *In vitro* testing identified multiple lead agents. Valproic acid was identified in this initial screen, as a known HDAC2 inhibitor and shown to interact with the nsp5 protease of SARS-CoV2. Unfortunately, *in vitro*, viral replication assays failed to demonstrate any antiviral activity of VPA at the doses tested. We conducted *in vitro* testing of VPA against coronavirus replication and confirmed those initial findings, though we did demonstrate antiviral activity at the doses bracketed around the known $IC_{50}$ for HDAC2 inhibition. However, that dosage range was determined to be too high for clinical use. Here we demonstrate that epidemiologic data support the case for VPA as an antiviral agent against coronavirus despite the predicted lack of activity from *in vitro*, high throughput screening assays. We investigate the screening methods, identify potential pitfalls in current screening methods, interrogate a surrogate-less pathogenic coronavirus (HCoV-229E) to dissect potential antiviral pathways, and identify a method to augment Valproic acid antiviral activity with the combination of VPA and DHA. These findings support a potentially powerful strategy for the cost-effective widespread preventive use of VPA/DHA against certain coronaviruses.

## Materials and methods

### Cell culture

MRC-5 cells (ATCC CCL-171) were grown in Eagle's Minimum Essential Medium (ATCC), and Vero cells (ATCC CCL-81) were grown in Dulbecco's Modified Eagle's Medium (ATCC),

both with 10% Fetal Bovine Serum (Gibco) and 1% PEN-STREP (Hyclone) at 37°C in a humidified chamber with 5% $CO_2$. Cell growth and cytotoxicity of Valproic acid (MP Biomedical LLC) were measured using an XTT Assay (ATCC) following manufacturer's instructions.

## Determination of valproic acid IC50

The $IC_{50}$ of VPA was determined in Vero and MRC-5 cells using three different assays. For antibody- based assay, Vero cells were grown to a monolayer in 5 out of 6 wells in a 6 well plate in DMEM with 2% FBS and infected with SARS-CoV-2 at MOI of 0.05 for one hour. One empty well was left out to serve as background from input SARS-CoV-2. Spent media was removed and 1500 μl of new media was added per well. Plate was incubated at 37°C for 24 hours. Cells were collected into a 2 mL Microfuge tube, vortexed, and spun down for 10 mins. Then 1 mL of supernatant was collected, while the pellet and remaining supernatant were discarded. Six treatment tubes were labeled accordingly, including media-only, Remdesivir, no drug (virus control), and three concentrations of VPA. Serial dilutions of the virus were prepared and added to a 48- well plate with Vero cells. After overnight incubation, plate was read using a standard spectrophotometer with an emission wavelength of 428nm.

For the luciferase assay, Vero cells were seeded at $0.3 \times 10^6$ cells/well in 12-well plates, incubated at 37°C with 5% CO2, and grown to ~90% confluency. Viral infection with SARS-CoV-2- NanoLuc (Promega) was performed by diluting stock virus to an MOI of 0.5/ml in DPBS. Then, 100 μL of the diluted stock was added to each well for a final MOI of 0.05/well. Cells and viruses were incubated at 37°C with 5% $CO_2$ for 1 hour, rocking the plates every 15 minutes. After this, the media was removed, 1 mL of prewarmed media with 1% FBS was added to each well, and the cells were incubated for 24 hours. The plate was then removed and allowed to cool to room temperature before 1 mL of prepared Nano-Glo Luciferase assay solution was added to each well and incubated for ~3min. After mixing by repeated pipetting, the lysate was transferred to a black luminance plate in quadruplicate and read using the luminescence endpoint protocol in a SpectraMax iD3 Multi-Mode Microplate Reader.

For the antibody-based dot blot assay, the MRC-5 cell line (Cat# CCL-171TM) was purchased fromATCC. The cells were cultured in the Eagle's Minimum Essential Medium (EMEM), including 10%fetal bovine serum (FBS) and Penicillin/Streptomycin. The cells were seeded at $1 \times 10^5$ per well of 12-well plates. For preincubation with Valproic acid, when the cells were at 70% confluence, Valproic acid at 3-fold serial dilution starting at 10mM was treated for 24 hour, and then the cells were inoculated with human coronavirus HCoV-229E (ATCC, Cat# VR-740) in 2% FBS EMEM for 1 hour at 37°C with 5% CO2. After an hour of viral incubation, the media was replaced with complete EMEM containing serially diluted Valproic acid at the concentration same as in pretreatment for an additional 48 hours. After 48 hours of drug treatment, the cell culture media were removed and washed with PBS, and then RIPA cell lysis buffer was added to the cells to lyse the cells and release the virus from the cells. The lysates were loaded onto a pre-wetted nylon membrane assembled in a 96-well Manifold unit. After the vacuum filtration, the membrane was incubated with 5% fat-free milk for 30 minutes and subsequently incubated with rabbit anti- HCoV-229E antibody (Cat# 40640-T62, SinoBiological) for 1 hour at room temperature and washed three times with TBS-T buffer, 10 minutes each. Then, the membrane was incubated with HRP-conjugated goat anti-rabbit IgG (H+L) secondary antibody (Cat# A16096, Invitrogen) for 1 hour at room temperature, washed three times with TBS-T buffer, 10 minutes each. The membrane was developed with Pierce ECL chemiluminescent substrate (Cat# 32106, ThermoFisher). The Un-Scan-It software was used for analyzing the intensity of the dot blot, and the GraphPad Prism software was used for statistics and graphs.

In the case of oligonucleotide detection of HCoV-229E, the membrane was incubated with anti HCoV-229E biotinylated-Oligo (5′ – CCACTCTCAACAGCAAATACATTTTCTGAATAA CCAACA constructed with 3' biotinylation) for 1 hour and washed with TBS-T 3x, 10 minutes each. The membrane was then incubated with Poly-HRP-Streptavidin (ThermoFisher Scientific, Cat# N200) in Poly-HRP dilution buffer (ThermoFisher Scientific, Cat# N500) for 30 minutes and washed with TBS-T 3x. The membrane was developed with ECL chemiluminescent substrate. Un-Scan-It software was used for analyzing the intensity of the dot blot, and GraphPad Prism 8.0 was used to plot the data.

## Histone deacetylase (HDAC)activity assay of Valproic acid

To measure Histone deacetylase (HDAC) activity assay of Valproic acid, the In-Situ HDAC Activity Assay Kit was obtained from LSBio (Cat# LS-K428-100). Vero cells were plated into a black wall/clear bottom 96 well plate, and when the cells were 70% confluent, Valproic acid at the 1:3 serially diluted concentrations starting at 100 mM were treated for 24 hours. The endpoint fluorometric measurement was performed at 368/442nm wavelengths of excitation/ emission following the protocol suggested by assay kit. For the p21 protein expression, Vero cells were treated with 0.5 mM Valproic acid at the designated time periods, and 30ug of the cell lysates were loaded into 12% SDS-PAGE gel and transferred to nitrocellulose membrane for western blot using p21 antibody (Upstate, Cat# 05–345) and GAPDH (Sigma, Cat# G9545).

## Impact of fatty acids and HDAC inhibitors on viral replication of HCoV-229E

MRC5 cells were grown to a confluence of 70% and incubated with each of the following polyunsaturated fatty acids: docosahexaenoic acid (DHA), eicosapentaenoic acid (EPA), linoleic acid (LA), or alpha-linoleic acid (αLA)- obtained from Cayman Chemicals. Briefly, for the single drug experiment, when the MRC5 cells were at 70% confluence, the virus was inoculated into the cells in 2% FBS media for an hour, and then the media was removed to replace it with complete media containing drugs at 1:3 serial dilutions starting at 100uM for 48 hours incubation. For the combined drugs experiment, the cells were pretreated with drugs for 24 hours, and then infected with the virus in 2% FBS media for an hour. The cells were incubated with HCoV-229E at an MOI of 0.1 for 1 hour. After an hour of virus inoculation, the media was removed and replaced with complete media containing drugs for an additional 48 hours of incubation. The cells were then harvested with Cell Lysis buffer (RIPA) as described above, and lysate dotted into a 96-well manifold with a nitrocellulose membrane for an antibody-based assay described above. The assay was performed with a 1:3 dilution of the selected fatty acid starting at 100 μM. The $IC_{50}$ of each fatty acid was calculated using GraphPad Prism, using sigmoidal 3-factor curve fitting. Positive control of no drug was included and used for 100% viral replication. A negative control without any drug or virus was used for baseline subtraction. The $IC_{50}$ from a curve was accepted if the following conditions were met: (1) the $R^2$ of the fitted curve was above 0.65, (2) the standard error (SE) of the $\log(IC_{50})$ corresponded to a fold change smaller than 8, (3) the SE of the slope parameter was smaller than 8. Initial statistical analysis for comparing two dose-response curves was performed by two-way ANOVA with Tukey HSD, and the two curves were considered different when $p < 0.05$. If two curves were considered different, a comparison of $IC_{50}$ was assessed using a statistical mixed-effect model to the $\log(IC_{50})$. Please note that in the case of Trichostatin A with and without DHA (Fig 4C), the $IC_{50}$ could not be accurately calculated, and no attempt was made for statistical comparison.

## RNA processing and RNAseq data analysis

MRC-5 cells were grown to ~70% confluency in complete media in 6-well plates, and then the culture media was replaced with serum-free media overnight. After this, the media was removed, and various drug combinations in complete media were added. For the time course study, 0.7 mM VPA was used, and cells were harvested every 24 hours for 96 hours. For the follow-up study, each well had one of the following conditions for 48 hours before harvesting: without HCoV-229E virus–media alone, 100 μM VPA, 500 μM VPA, 25 μM DHA, 100 μM VPA with 25 μM DHA, and 500 μM VPA with 25 μM DHA. With HCoV-229E virus–media alone, 500 μM VPA, 25 μM DHA, 100 μM VPA with 25 μM DHA, 500 μM VPA with 25 μM DHA, and 500 μM VPA with 25 μM DHA pre-incubated for 24 hours before viral infection. Cells were harvested by washing the cells twice with phosphate-buffered saline (ATCC), then trypsinizing the cells, neutralizing the trypsin with complete media, and centrifuging at 1500 rpm for 5 min. After removing the media, the cells were washed with PBS and spun down again. Finally, 500 μL of Trizol (Sigma) was added, and the standard protocol was followed to isolate RNA. RNA concentration was determined using UV-vis spectrometry at 260/280 nm in a Spectra-Max i3 plate reader (Molecular Devices).

Library preparation was completed on approximately 500 ng of total RNA using a standard workflow using the KAPA Stranded RNA-Seq Kit with RiboErase (KAPA Biosystems). Briefly, rRNA was depleted, and then the remaining RNA was fragmented using divalent cations under elevated temperature and magnesium. The fragmented RNA was copied into double-stranded cDNA using random primers. Following this, adapters were ligated to the ends of the cDNA, and the product was amplified using PCR, which further purified and enriched the sequences for a final RNA-sequencing library. The libraries were quantified using a Qubit (ThermoFisher) and Bioanalyzer (Agilent). The libraries were pooled to 20 nM concentrations for sequencing. Sequences were read on a NextSeq 500 (Illumina) with 75 bp paired-end reads with an average of 35 million reads per sample. Raw data were demultiplexed to generate FASTQ files for analysis. Raw FASTQ files were uploaded to CLC Genomics Workbench 21 (Qiagen) for processing and initial analysis.

The provided RNA sequencing workflow was used to first quality control the raw data and removeany poor reads. Then the sequences were trimmed to remove the adapters. The sequences were then mapped to the homo sapiens hg38 reference genome, and the gene count was generated. Gene counts were normalized to RPKM for differential expression analysis. Each treatment condition was compared to its respective control (media alone, with or without HCoV-229E virus) to examine changes in gene expression due to drug treatment. For differential expression analysis, the threshold for significance was set at over 1.5 fold change in expression and FDR p- value of less than 0.05. Differential gene expression was compared with CLC genomics set to pairwise expression comparing RPKM of one condition to the sham infected sham-treated MRC5 cells. The data was segregated into two groups. The group with the drug only was compared to MRC5 cells without the drug. The group with drug and virus was compared to MRC5 cells infected with the HCoV-229E virus but no drug. Plots were made into NCCS comparing Log2(fold expression) between the two conditions in a 2-D scatterplot for the top 2000 differentially expressed genes. The gene expression data were processed through iDEP 1.9 and k-means cluster for 6 group clusters. The clusters were color-coded when plotted on the 2-D scatterplots to permit visualization of the impact on cluster groups by treatment.

Additionally, the % of Viral compared to total cellular RNA was determined by mapping the reads to the HCoV-229E viral reference genome (NCBI), and the total counts of viral RNA were determined and plotted as a percentage of total RNA sequences.

## Time course data analysis

After differential expression analysis, the total number of significantly up and down-regulated genes were plotted in GraphPad Prism. The gene list was then compared to the approximately 300 genes that interacted with SARS-CoV-2 genes per Gordon *et al.* [10] to determine which target genes could affect the virus. The $\log_2$(fold change) was plotted vs. the $-\log_{10}$(p-value) to create a volcano plot in NCSS software. The genes that met the significance criteria were mapped out in Cytoscape to visualize the virus- host protein interactions. Viral proteins were categorized relative to viral activity as viral assembly, replication, or pathogenicity. Western blot on select targets demonstrated host proteins to confirm the RNA-sequencing results at the protein level.

## VPA and DHA RNAseq study data analysis

Mapped data was uploaded to iDep v.94 [14] and analyzed using its built-in algorithms and workflow. In summary, total differential gene expression was mapped in a heat map for the top 12,000 differentially expressed genes with at least 1.5 fold change relative to control and a p- value of 0.05. K-means clusters were generated for an appropriate number of clusters relative to the data and then plotted as a scatter plot in NCSS software. Principle components analysis (PCA) was also completed to examine how the samples were grouped in CLC Genomics (Qiagen, Inc). Differential expression analysis was done with DEG2, an FDR threshold p-value less than 0.05, and at least 1.5-fold change. Differential gene expression patterns were examined using a Venn diagram of unique and shared genes across samples. Finally, gene ontology (GO) analysis was performed using all three categories: Biological Processes, Cellular Components, and Molecular Function. The GO analysis was further refined using PGSEA [15].

## Western blots

MRC-5 cells were grown to ~70% confluency and then treated with Valproic acid (MP Biomedical LLC) at concentrations between 0 and 5 mM for 24 or 48 hours. Protein was isolated by lysing cells in radioimmunoprecipitation buffer (Santa Cruz Biochemistry) for 1 hour, followed by centrifugation at 10,000 x g for 15 minutes at 4˚C. Protein concentration was determined using the Pierce BCA protein assay kit. For immunoblotting, proteins were separated using SDS-PAGE and transferred to nitrocellulose membranes. The membranes were blocked with 5% nonfat dry milk in Tris-buffered saline and then incubated with anti-PCNT (ab99341, Abcam), anti-DNMT1 (ab188453, Abcam), anti-BRD2 (ab139690, Abcam), anti-HMOX1 (ab52947, Abcam), and anti- GAPDH (G9545, Sigma). The membranes were incubated with appropriate horseradish peroxidase-conjugated secondary antibodies, and then bands were visualized by enhanced chemiluminescence. Densitometric analysis was performed using UnScan-It 7.0 software (Silk Scientific).

## Optum dataset analysis

EMR data from the Optum data set representing more than 3 million patients were collected between the first quarter of 2020 to the second quarter of 2021 for all patients undergoing COVID-19 testing. The data are analyzed as a case-control for factors associated with COVID-19 infection and retrospective cohort study amongst COVID-19 cases to study outcomes. All patients were tested for COVID-19 by a nucleic acids method. Variables used for adjustment and matching were gender, age, race, ethnicity, insurance, vaccine status, cancer, cancer treatment, congestive heart failure (CHF), chronic obstructive pulmonary disease (COPD), renal disease, diabetes mellitus (DM), region of the country where the patient was tested, and the

calendar trimester when the patient was tested. The central hypothesis tested is the association with the detection of COVID-19 by nucleic acid testing in patients with documentation of Valproic acid use in their medication list. The null hypothesis (H$_0$) is that SARS-CoV2- test positivity has no association with VPA therapy. The secondary study is a retrospective cohort study conducted with multivariate analysis, using Logistic regression for the categorical variables as listed in Table 1A and 1B. The analysis was performed using R software and NCSS (version 2021). The null hypothesis (H$_0$) for this portion of the analysis is that SARS-CoV2 test positivity has the same odds ratio for 30-day all-cause ER admission regardless of VPA use and that SARS-CoV2 test positivity has the same odds ratio for 30-day all-cause hospital admission regardless of VPA use. Frequency distribution of the patient demographics and prior medical history was performed for gender, race, ethnicity, geography, and comorbidities. Notably, the patients treated with VPA had higher levels of acquired immunodeficiency syndrome (AIDS), CHF, COPD, Cerebral vascular disease, dementia, diabetes, prior MI, and peripheral vascular disease. As these conditions are known to predispose to a higher risk of morbidity from SARS-CoV2, a fully adjusted regression model and a 1:1 nearest neighbor propensity score-based matching model were created in which patients without listed VPA use were balanced for comorbidities including AIDS, CHF, CPOD, CVD, dementia, diabetes, prior MI and PVD. Comparisons between the VPA and non- VPA groups were performed using logistic regression.

## Valproic acid levels in a national data set

All serum VPA levels tested during the third week of June 2021 were provided kindly by LabCorp. Data were analyzed by gender, with similar distributions, and separated by quartiles for comparisons. No attempt was made to match patients between the LapCorp cohort and the Optum cohort, as both groups were fully de-identified. No information on the dosing of Valproic acid is known for the patients studied from either cohort (Optum or LabCorp).

## NGS for drug inhibition of HCoV-229E replication

Differential gene expression analysis was done in two groups. The first group (group 1) were MRC5 cells in which the cells were treated with VPA, DHA, VPA+DHA, or control (no drug). The reference control for the differential expression was the expression of MRC5 without a drug. The second group (group 2) were MRC5 cells in which the cells were treated with the HCoV-229E virus, then drug for 48 hours, and harvested for RNA as described in the methods section for RNASeq. The control group was MRC5 cells treated with HCoV-229E virus without drug, and an additional set of cells was pre-incubated with 0.5 mM VPA + 25 uM DHA for 24 hrs prior to HCoV-229E incubation (the most robust condition for inhibition of viral replication). The group 2 cells had their gene expression referenced to the MRC5 incubated with virus only (no drug). The RNA was run for RNASeq as described above and then analyzed relative to each group control. Shared genes for the conditions, including DHA, VPA, or DHA +VPA, were assessed for differential gene expression in both the drug-treated and the drug-treated/virus-treated groups. Differential gene expression was then assessed under VPA, DHA, and DHA+VPA conditions and screened for p<0.05, with a gene expression cutoff of 1.5 fold relative to control MRC5 or control MRC5 infected with HCoV- 229E. Hierarchical clustering was performed with dual dendrogram heat maps plotting absolute fold gene expression with Z-scale normalization (to provide linear clustering but symmetric expression around 0 for induction and repression).

**Table 1. Characteristics of patients in the Optum dataset listed as having been prescribed VPA in the retrospective cohort of patients tested for COVID-19.** A. Demographics and collection period. B. Vaccine status and comorbidities.

| | | A. | | | |
|---|---|---|---|---|---|
| | | **COVID-19-ve (Controls)** | | **COVID-19+ve (Cases)** | |
| | | **(N = 2,717,216)** | | **(N = 405,234)** | |
| **Demographics** | | **Gender** | | | |
| | Male | 1,098,964 | 40.4% | 172,434 | 42.6% |
| | Female | 1,618,252 | 59.6% | 232,800 | 57.4% |
| | | **Age** | | | |
| | Mean (SD) | 51.1 | (18) | 48.7 | (17.8) |
| | Median [Min, Max] | 52.0 | [19.0, 88.0] | 48.0 | [19.0, 88.0] |
| | | **Race** | | | |
| | White | 2,118,952 | 78.00% | 295,962 | 73.00% |
| | Black | 314,845 | 11.60% | 53,660 | 13.20% |
| | Asian | 51,944 | 1.90% | 7,696 | 1.90% |
| | Other/Unknown | 231,475 | 8.50% | 47,916 | 11.80% |
| | | **Ethnicity** | | | |
| | Not Hispanic | 2,340,650 | 86.10% | 335,287 | 82.70% |
| | Hispanic | 160,206 | 5.90% | 44,519 | 11.00% |
| | Unknown | 216,360 | 8.00% | 25,428 | 6.30% |
| | | **Geographical Region** | | | |
| | Midwest | 1,328,867 | 48.90% | 218,018 | 53.80% |
| | Northeast | 578,898 | 21.30% | 78,045 | 19.30% |
| | South | 419,903 | 15.50% | 65,757 | 16.20% |
| | West | 280,403 | 10.30% | 28,369 | 7.00% |
| | Unknown | 109,145 | 4.00% | 15,045 | 3.70% |
| | | **Insurance Status** | | | |
| | Commercial | 1,494,840 | 55.00% | 246,755 | 60.90% |
| | Medicare | 672,380 | 24.70% | 79,948 | 19.70% |
| | Medicaid | 344,070 | 12.70% | 50,263 | 12.40% |
| | Uninsured | 52,117 | 1.90% | 9,685 | 2.40% |
| | Unknown | 153,809 | 5.70% | 18,583 | 4.60% |
| **Data collection Period** | | **Calendar Quarter of COVID-19 Dx** | | | |
| | 2020.Q1 | 37,102 | 1.40% | 8,569 | 2.10% |
| | 2020.Q2 | 555,859 | 20.50% | 46,966 | 11.60% |
| | 2020.Q3 | 780,776 | 28.70% | 64,380 | 15.90% |
| | 2020.Q4 | 663,889 | 24.40% | 182,503 | 45.00% |
| | 2021.Q1 | 416,438 | 15.30% | 76,242 | 18.80% |
| | 2021.Q2 | 263,152 | 9.70% | 26,574 | 6.60% |
| | | B. | | | |
| | | **COVID-19-ve (Controls)** | | **COVID-19+ve (Cases)** | |
| | | **(N = 2,717,216)** | | **(N = 405,234)** | |
| **Vaccine Status** | | **At least 1 prior dose of COVID-19 vaccine** | | | |
| | No | 2,640,896 | 97.20% | 402,942 | 99.40% |
| | Yes | 76,320 | 2.80% | 2,292 | 0.60% |

(*Continued*)

**Table 1.** (Continued)

| Existing Comorbidities | | | | | |
|---|---|---|---|---|---|
| | **Cancer** | | | | |
| No | | 2,395,620 | 88.20% | 372,790 | 92.00% |
| Yes | | 321,596 | 11.80% | 32,444 | 8.00% |
| | **Metastasis** | | | | |
| No | | 2,633,185 | 96.90% | 397,913 | 98.20% |
| Yes | | 84,031 | 3.10% | 7,321 | 1.80% |
| | **Cancer Active (Rx within past 90 days)** | | | | |
| No | | 2,683,329 | 98.80% | 400,789 | 98.90% |
| Yes | | 33,887 | 1.20% | 4,445 | 1.10% |
| | **Myocardial infarction** | | | | |
| No | | 2,500,420 | 92.00% | 376,827 | 93.00% |
| Yes | | 216,796 | 8.00% | 28,407 | 7.00% |
| | **Congestive Heart Failure** | | | | |
| No | | 2,422,704 | 89.20% | 367,278 | 90.60% |
| Yes | | 294,512 | 10.80% | 37,956 | 9.40% |
| | **Chronic Obstructive Pulmonary Disease** | | | | |
| No | | 1,834,429 | 67.50% | 284,079 | 70.10% |
| Yes | | 882,787 | 32.50% | 121,155 | 29.90% |
| | **Diabetes mellitus** | | | | |
| No | | 2,165,916 | 79.70% | 317,140 | 78.30% |
| Yes | | 551,300 | 20.30% | 88,094 | 21.70% |
| | **Diabetes mellitus with end organ damage** | | | | |
| No | | 2,455,469 | 90.40% | 364,830 | 90.00% |
| Yes | | 261,747 | 9.60% | 40,404 | 10.00% |
| | **Liver Disease** | | | | |
| No | | 2,397,139 | 88.20% | 363,925 | 89.80% |
| Yes | | 320,077 | 11.80% | 41,309 | 10.20% |
| | **Moderate to severe Renal Disease** | | | | |
| No | | 2,429,377 | 89.40% | 363,792 | 89.80% |
| Yes | | 287,839 | 10.60% | 41,442 | 10.20% |

## Treatment of MRC5 cells with DHA and VPA prior to infection with SARS-CoV2 to inhibit replication

In our initial experiment to determine if pretreatment of MRC-5 cells with drug would inhibit replication of SARS-CoV2 virus we grew MRC-5 cells in the presence of DHA (25 uM), VPA (0.5 mM), or DHA and VPA (25 uM and 0.5 mM respectively) or media not containing drug, for 4 days before plating 300,000 pretreated cells into 6 well plates in the appropriate drug containing media. The following day, SARS-Cov2 virus was added to each well at a MOI of 0.1, with no virus acting as a control. Media and virus was removed after 1 hour and was replaced with media containing the appropriate drug. After incubating the cells for 24 hours, media was removed and RNA harvested from the cells using Trizol. RNA was analysed for SARS CoV2 RNA by real time RT- PCR. This experiment was then repeated, but with the addition of a 3 day pretreatment with the drugs before viral infection and performed in 12 well plates with 80,000 cells per well. Total RNA was extracted via Trizol. Libraries were generated from NEB Directional RNA library preparation guide (NEBNEXT rRNA depletion kit v2 human/mouse, New England Biolabs, Ipswich, MA). Strand specificity was obtained by dUTP incorporation

during second strand synthesis followed by poly- A end repair and adapter ligation. Subsequently, Total RNA-Seq libraries were quantified and pooled for paired-end sequencing via NextSeq 150HO. This work was performed by the Genome Sequencing Facility with the support of NIH-NCI P30 CA054174 (Cancer Center at UT Health San Antonio), NIH Shared Instrument grant 1S10OD021805-01 (S10 grant), and CPRIT Core Facility Award (RP160732).

## Results

### Case-control analysis of the optum dataset

The Optum dataset collected information from more than 3 million patients tested for COVID-19 using Nucleic Acid testing from the First quarter of 2020 until the second quarter of 2021. As shown in Table 1, more than 400,000 patients tested positive for COVID-19, and 2.7M patients tested negative for a net positivity rate of 14.9%, with a similar distribution between males and females—though females were tested significantly more often than males (1.46 fold). The age distributions between males and females were comparable between COVID+ and COVID- patients. At the time of this COVID-19 testing, significant geographic differences were not observed between COVID+ and COVID- patients. Race and ethnicity patterns did demonstrate higher percentages of Black and Hispanic patients in the COVID+ cohort, consistent with other previously reported statistics [16–19]. Table 2 shows the distribution of positive and negative patients over the study period, from the first quarter of 2020, until the second quarter of 2021. The peak of the test positivity was in the fourth quarter of 2020, representing 45% of all of the positive cases in the study period. The insurance status between COVID+ and COVID- patients was similar over the study period. Importantly, >97% of the COVID+ and COVID- patients had no prior immunization at the time of testing. This finding permits an evaluation of the potential impact of VPA use without the confounding impact of prior immunization. Table 3 (top) shows the odds ratios for univariate and multivariate analyses of the likelihood of test positivity when patients have been "exposed" to a VPA prescription, as determined by their active medications list. Actual serum VPA measurements were unavailable for this patient cohort during the COVID-19 testing.

As a control, a sampling of all the VPA levels collected through LabCorp during two weeks in June 2021 is shown in (S1 Fig). The distribution of serum VPA levels demonstrates a mild platykurtic near-normal distribution of values with skewness of -0.009953 and kurtosis of–0.3546, with a mean and median of 56.7 μg/mL and 59.0 μg/mL, respectively. S1 Fig shows that fewer than 25% of the patients tested had a serum level of 80 μg/mL (~0.55 mM). Most patients on VPA for a seizure disorder have target concentrations of 50–100 μg/mL, with enhanced toxicity found at levels > 125 μg/mL. If less than 50% of all patients on VPA have achieved therapeutic levels (assuming they are the same between seizure disorder and COVID+ protection), then the OR found in Table 3 may underestimate the protective effect by more than two-fold. The association of VPA use with a decreased rate of positive COVID-19 testing does not

**Table 2. Odds ratio (OR) of "exposure to VPA prescription" in patients diagnosed with COVID-19 in the retrospective cohort of patients tested for COVID-19.**

|  | OR | 95% CI |
|---|---|---|
| Crude | 0.83 | 0.81–0.86 |
| Adjusted | 0.88 | 0.85–0.91 |
| Exact Matched | 0.75 | 0.72–0.78 |

**Table 3. Characteristics of patients prescribed or not prescribed VPA in the retrospectivecohort of COVID-19 infected patients.** A. Demographics and collection period. B. Vaccine status and comorbidities.

| | | A. | | | |
|---|---|---|---|---|---|
| | | **COVID +ve Patients** | | | |
| | | **No VPA Treatment (Controls)** | | **VPA Treatment (Cases)** | |
| | | **(N = 261,349)** | | **(N = 2,927)** | |
| **Demographics** | | **Gender** | | | |
| | Male | 102,751 | 39.3% | 1,310 | 44.8% |
| | Female | 158,598 | 60.7% | 1,617 | 55.2% |
| | | **Age** | | | |
| | Mean (SD) | 49.7 | (17.4) | 50.0 | (16.7) |
| | Median [Min, Max] | 50.0 | [19.0, 88.0] | 50.0 | [19.0, 88.0] |
| | | **Race** | | | |
| | White | 192,724 | 73.7% | 2,210 | 75.5% |
| | Black | 34,263 | 13.1% | 438 | 15.0% |
| | Asian | 4,989 | 1.9% | 36 | 1.2% |
| | Other/Unknown | 29,373 | 11.2% | 243 | 8.3% |
| | | **Ethnicity** | | | |
| | Not Hispanic | 218,142 | 83.5% | 2,568 | 87.7% |
| | Hispanic | 28,111 | 10.8% | 262 | 9.0% |
| | Unknown | 15,096 | 5.8% | 97 | 3.3% |
| | | **Geographical Region** | | | |
| | Midwest | 140,161 | 53.6% | 1,636 | 55.9% |
| | Northeast | 55,556 | 21.3% | 616 | 21.0% |
| | South | 39,174 | 15.0% | 350 | 12.0% |
| | West | 17,221 | 6.6% | 165 | 5.6% |
| | Unknown | 9,237 | 3.5% | 160 | 5.5% |
| | | **Insurance Status** | | | |
| | Commercial | 159,606 | 61.1% | 985 | 33.7% |
| | Medicare | 53,547 | 20.5% | 1,063 | 36.3% |
| | Medicaid | 32,138 | 12.3% | 696 | 23.8% |
| | Uninsured | 4,539 | 1.7% | 28 | 1.0% |
| | Unknown | 11,519 | 4.4% | 155 | 5.3% |
| **Data collection Period** | | **Calendar Quarter of COVID-19 Dx** | | | |
| | 2020.Q1 | 6,605 | 2.5% | 61 | 2.1% |
| | 2020.Q2 | 34,241 | 13.1% | 504 | 17.2% |
| | 2020.Q3 | 44,913 | 17.2% | 475 | 16.2% |
| | 2020.Q4 | 117,902 | 45.1% | 1,206 | 41.2% |
| | 2021.Q1 | 48,783 | 18.7% | 553 | 18.9% |
| | 2021.Q2 | 8,905 | 3.4% | 128 | 4.4% |
| | | B. | | | |
| | | **COVID +ve Patients** | | | |
| | | **No VPA Treatment (Controls)** | | **VPA Treatment (Cases)** | |
| | | **(N = 261,349)** | | **(N = 2,927)** | |
| **Vaccine Status** | | **At least 1 prior dose of COVID-19 vaccine** | | | |
| | No | 259,930 | 99.5% | 2,904 | 99.2% |
| | Yes | 1,419 | 0.5% | 23 | 0.8% |

(*Continued*)

**Table 3.** (Continued)

| Existing Comorbidities | | | | | |
|---|---|---|---|---|---|
| **Cancer** | | | | | |
| No | 236,851 | 90.6% | 2,627 | 89.8% |
| Yes | 24,498 | 9.4% | 300 | 10.2% |
| **Metastasis** | | | | | |
| No | 255,749 | 97.9% | 2,865 | 97.9% |
| Yes | 5,600 | 2.1% | 62 | 2.1% |
| **Cancer Active (Rx within past 90 days)** | | | | | |
| No | 257,882 | 98.7% | 2,886 | 98.6% |
| Yes | 3,467 | 1.3% | 41 | 1.4% |
| **Myocardial infarction** | | | | | |
| No | 242,055 | 92.6% | 2,530 | 86.4% |
| Yes | 19,294 | 7.4% | 397 | 13.6% |
| **Congestive Heart Failure** | | | | | |
| No | 235,047 | 89.9% | 2,367 | 80.9% |
| Yes | 26,302 | 10.1% | 560 | 19.1% |
| **Chronic Obstructive Pulmonary Disease** | | | | | |
| No | 175,311 | 67.1% | 1,428 | 48.8% |
| Yes | 86,038 | 32.9% | 1,499 | 51.2% |
| **Diabetes mellitus** | | | | | |
| No | 198,023 | 75.8% | 1,931 | 66.0% |
| Yes | 63,326 | 24.2% | 996 | 34.0% |
| **Diabetes mellitus with end organ damage** | | | | | |
| No | 231,783 | 88.7% | 2,377 | 81.2% |
| Yes | 29,566 | 11.3% | 550 | 18.8% |
| **Liver Disease** | | | | | |
| No | 229,846 | 87.9% | 2,339 | 79.9% |
| Yes | 31,503 | 12.1% | 588 | 20.1% |
| **Moderate to severe Renal Disease** | | | | | |
| No | 232,496 | 89.0% | 2,362 | 80.7% |
| Yes | 28,853 | 11.0% | 565 | 19.3% |

necessarily demonstrate causality. However, these findings are consistent with the possible role of VPA in diminishing the rate of COVID-19 and the severity of the disease among those who test positive. Also shown in Table 3 is the distribution of patients testing positive for COVID-19 amounts in the VPA+ and VPA- groups for Age, Race, and Geography. VPA+ and VPA- patients had similar distributions for these three variables.

Logistic regression of a univariate (unadjusted) comparison of patients "exposed" to VPA compared to VPA-negative patients shows no protective effect against Emergency Room Visits; however, when either a propensity-matched model or a multivariate-adjusted model is applied, controlling for comorbidities such as diabetes, heart disease, congestive heart failure, and chronic obstructive pulmonary disease, then significant reductions were seen of 12–15% in ER Visits, 17–45% reduction in Hospital Admission, 33–39% reduction in mechanical ventilation, and 14–16% reduction in ICU admissions as shown in Table 4. These findings suggest that patients who actively take VPA are less likely to develop COVID-19, and when they do develop this infection, they are less likely to requirean Emergency Department visit, hospitalization, ICU admission, or mechanical ventilation.

**Table 4. Clinical progression of COVID-19 in patients prescribed or not prescribed VPA in the COVID-19 infected cohort.**

|  | No (Unexposed) (N = 261,349) | | Yes (Exposed) (N = 2,927) | | Unadjusted | Adjusted | Propensity Matched |
|---|---|---|---|---|---|---|---|
| **Emergency Room Admission** | | | | | | | |
| No | 206,300 | 78.9% | 2,230 | 76.2% | 1.17 (1.07–1.28) | 0.85 (0.77–0.92) | 0.88 (0.78–0.99) |
| Yes | 55,049 | 21.1% | 697 | 23.8% | | | |
| **Inpatient Admission** | | | | | | | |
| No | 232,099 | 88.8% | 2,637 | 90.1% | 0.87 (0.77–0.98) | 0.55 (0.48–0.62) | 0.62 (0.53–0.73) |
| Yes | 29,250 | 11.2% | 290 | 9.9% | | | |
| **Mechanical Ventilation** | | | | | | | |
| No | 257,764 | 98.6% | 2,889 | 98.7% | 0.94 (0.66–1.30) | 0.61 (0.44–0.85) | 0.67 (0.44–1.02) |
| Yes | 3,585 | 1.4% | 38 | 1.3% | | | |
| **ICU Admission** | | | | | | | |
| No | 256,966 | 98.3% | 2,865 | 97.9% | 1.26 (0.97–1.63) | 0.86 (0.66–1.11) | 0.84 (0.60–1.19) |
| Yes | 4,383 | 1.7% | 62 | 2.1% | | | |

## VPA Inhibition of SARS-CoV2 viral replication

The nonstructural protein 5 (Nsp5) of SARS-CoV2 interacts directly with HDAC2 [17], and agents which inhibit HDAC2 were initially predicted to potentiate viral replication [13, 17]. However, subsequent models predicted that inhibitors of HDAC2 might be effective antiviral agents against SARS-CoV2 [20, 21]. As VPA is already a widely used therapeutic with HDAC2 inhibitory activity, multiple groups proposed its use against SARS-CoV2 [22, 23]. However, viral inhibition assays failed to demonstrate inhibition of viral replication at the tested doses [10]. Since our epidemiological data support a putative role for VPA inhibition of SARS-CoV2 activity, we interrogated the *in vitro* activity of VPA in more detail. To determine the optimal dosing for an effect of VPA in viral replication, we measured the $IC_{50}$ of VPA in directly inhibiting HDAC2 activity in Vero cells (S1A Fig) and determined it to be at least 2.5 mM-a level almost threefold higher than accepted toxicity thresholds in humans. As p21 induction is a common consequence of HDACi, we evaluated the time course for induction and found that maximal induction occurs at 72 hours, with the earliest induction seen at 24 hours(S1B Fig). Since the initial models of VPA inhibition of SARS-CoV2 predicted a direct inhibition of HDAC2, we bracketed our viral inhibition assays around this $IC_{50}$ (Fig 1A–1D). Viral inhibition was measured by an antibody-based assay (Fig 1A), Firefly Luciferase assay (Fig 1B), and by oligonucleotide hybridization-based assay (Fig 1C and 1D) with similar results for both SARS-CoV2 (A and B) and HCoV-229E (C). However, preincubation with VPA for 24 or 48 hours reduces the $IC_{50}$ for inhibition of viral replication by more than 3-fold (Fig 1D), bringing the dose range for VPA within therapeutic limits. As SARS-CoV2 is pathogenic and requires specialized facilities for manipulation, we chose to further study the impact of VPA on viral replication on the related alpha-coronavirus HCoV-229E [24], thought to be responsible for many cases of the "common cold" [25]. HDAC inhibition is an effective mechanism for altering gene expression patterns towards a more differentiated state and has been used as an approach for several malignancies [5, 6]. When used as a cancer therapeutic, the effect of VPA is also delayed, as it takes time for gene expression patterns to manifest cellular changes [5]. Chronic administration of VPA, for instance, in prostate cancer cell lines can cause significant cellular toxicity and alterations in androgen receptor levels at much lower doses than observed during acute administration [6].

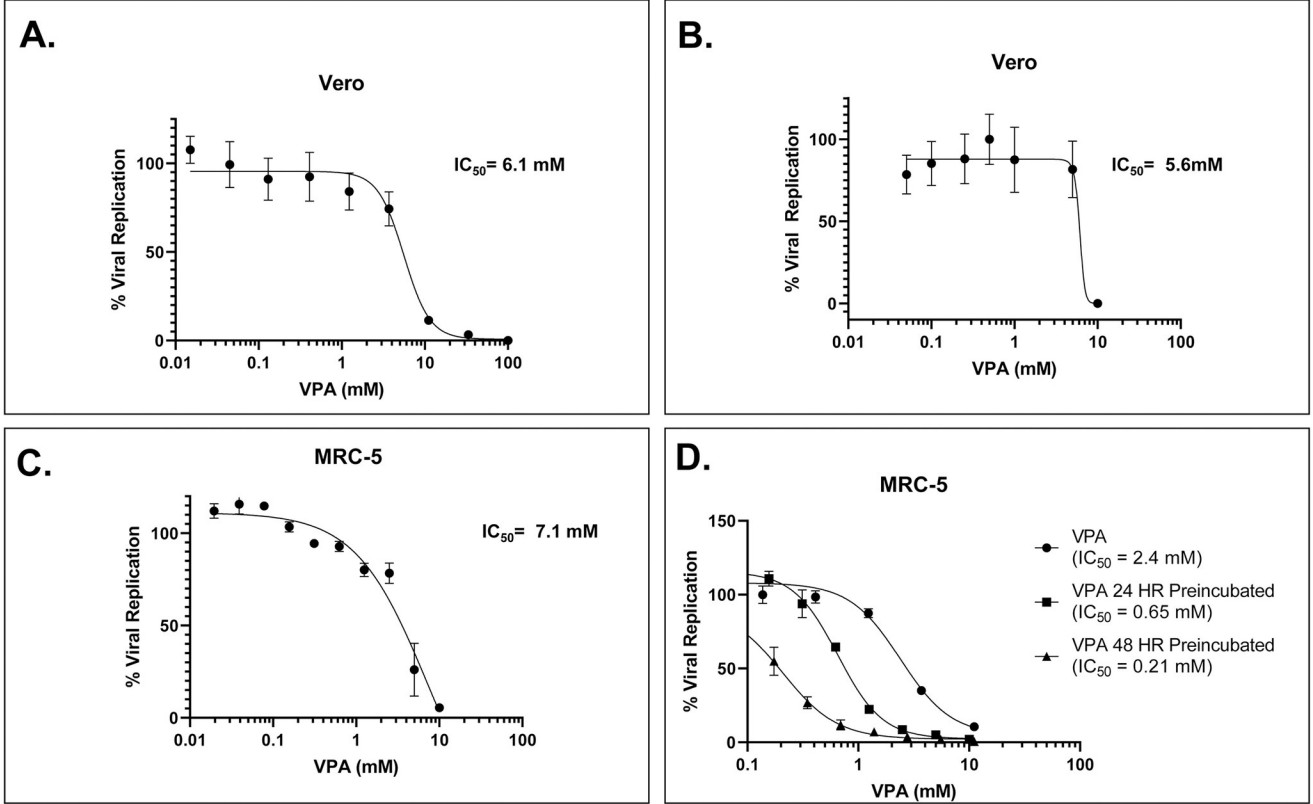

**Fig 1. Viral inhibition assays around the IC50 of VPA by direct inhibition of HDAC2.** (A.)Antibody-based assay infected with SARS-CoV2. (B.) Firefly luciferase assay infected with SARS-CoV2 (C.) Oligonucleotide hybridization-based assay infected with HCoV-229E. (D.) Oligonucleotide hybridization-based assay infected with HCoV-229E with/without VPA preincubation.

Gordon *et al.* studied the host cellular genes interacting with the various proteins produced by SARS-CoV2 and identified more than 300 human genes that directly bind to and interact with SARS-CoV proteins [10]. In order to help predict the impact of VPA gene expression changes, we performed RNAseq on the hCoV-229E permissive MRC5 cells treated with VPA for 24 and 48 hours and assessed the impact on the expression of the "Gordon" gene set. Fig 2A and 2B shows volcano plots of the 300 gene set, demonstrating that at 24 and 48 hours, there are changes in gene expression with both down and up- regulated genes. The overall impact is more down-regulated than up-regulated genes, with a significant difference in the down-regulation of genes involved in viral replication, as defined by Gordon *et al.* [10]. Both viral pathogenicity and replication genes are down-regulated preferentially with VPA treatment. Confirmation of the gene expression was performed by Western blot for PCNT, DNMT1, BRD2, and HMOX1 at 24 and 48 hours, consistent with the gene expression data (Fig 2C). Overall gene expression with VPA treatment demonstrated significant differences in the number of differentially expressed genes, with a predominance of up-regulated genes at 24 and 48 hours (Fig 2D).

Published efforts to test VPA *in vitro* as a potential antiviral agent were never performed with sufficient time for VPA to induce target genes, and in many assays, the cells were dead from viral replication before induction of HDAC2-regulated genes could even begin. In our clinical cohort, the ongoing use of VPA in patients with seizure disorders may have prevented or reduced the impact of COVID-19 infection, as VPA-targeted genes were already induced.

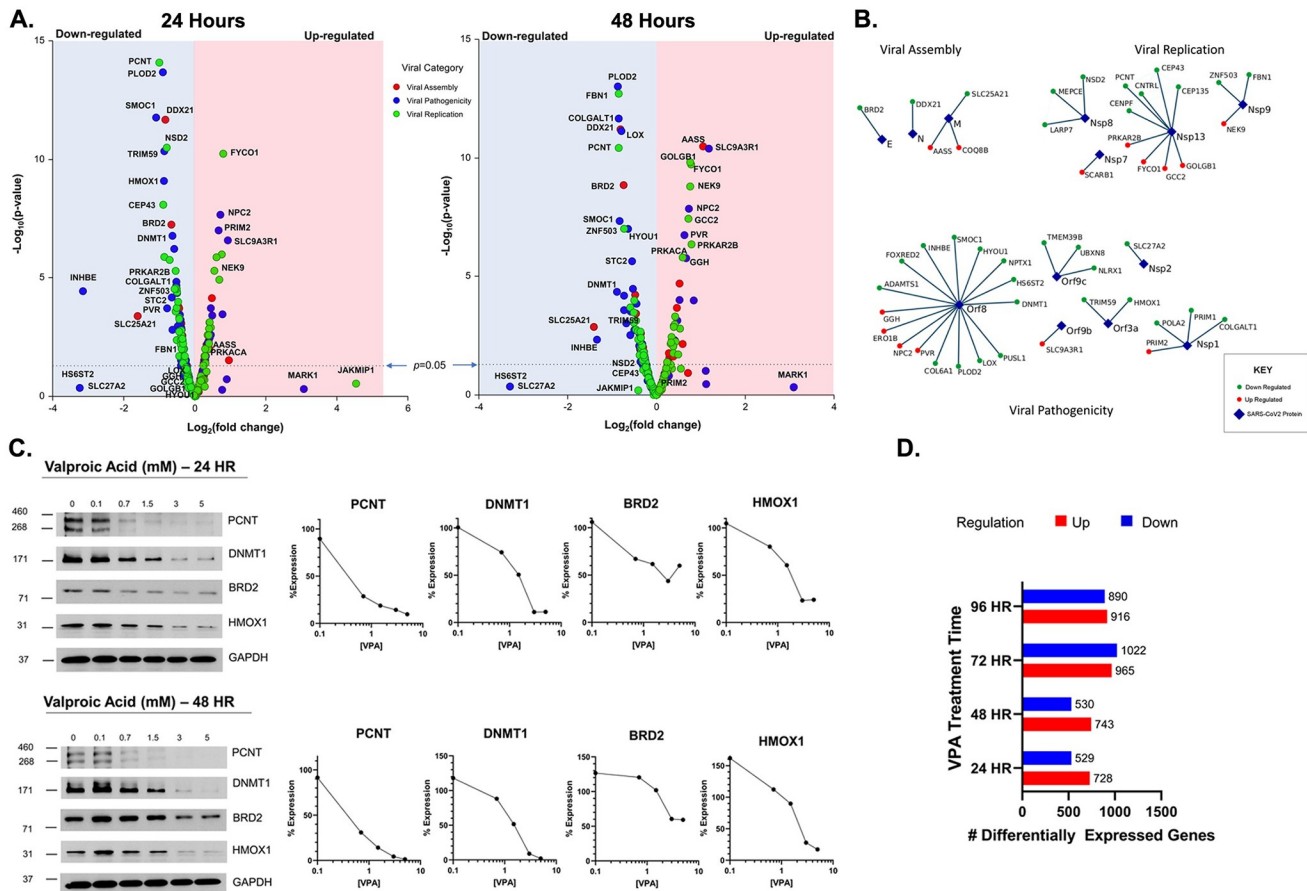

**Fig 2. Host cellular genes interactions with the SARS-CoV2 proteins.** (A.): Volcano plot for gene expression levels of selected 300 Gordon gene setat 24 hours (left) and 48 hours (right). (B.): Virus-host protein interaction map of the Gordon gene sets that met significance criteria for Viral Assembly, replication and pathogenicity. (C.) Protein levels measured by western blot for selected gene sets PCNT, DNMT1, BRD2, and HMOX1 at 24 hours (top) and 48 hours (bottom). (D.) Differentially expressed genes upon VPA treatment for 24, 48, 72 and 96 hours.

## Effect of polyunsaturated fatty acids (PUFAs) on HCoV-229E viral replication

Previous studies have demonstrated that the omega-3 fatty acid docosahexaenoic acid (DHA) can lessen the hepatic toxicity of VPA [25, 26], provide neuroprotection against VPA toxicity in rat fetuses [27, 28], augment antiseizure efficacy [29] as well as improve clinical efficacy in borderline personality disorder in some patients [30]. Additionally, some early studies demonstrated a potential fatty acid binding site in the Spike protein of SARS-CoV2, which is conserved in the HCoV- 229E virus and can be inhibited by linoleic acid [31–34]. Hence, we tested polyunsaturated fatty acids (PUFAs) in models of HCoV-229E viral inhibition. We tested the omega-3 fatty acids EPA (eicosapentaenoic acid), ALA (alpha-linolenic acid), and DHA (docosahexaenoic acid) as well as the omega-6 fatty acid LA (linoleic acid) for their effect on HCoV-229E viral replication in MRC5 cells. Cells were incubated with HCoV-229E for one hour at an MOI of 0.1, and media containing varying amounts of the PUFAs were added. The cells were incubated for another 48 hours before harvesting and assessing for viral replication using an antibody-based assay. As shown in Fig 3A, LA had a substantial inhibitory effect on HcoV-229E, as previously reported [31–34]. Surprisingly, DHA (Fig 3B), and EPA (Fig 3C)

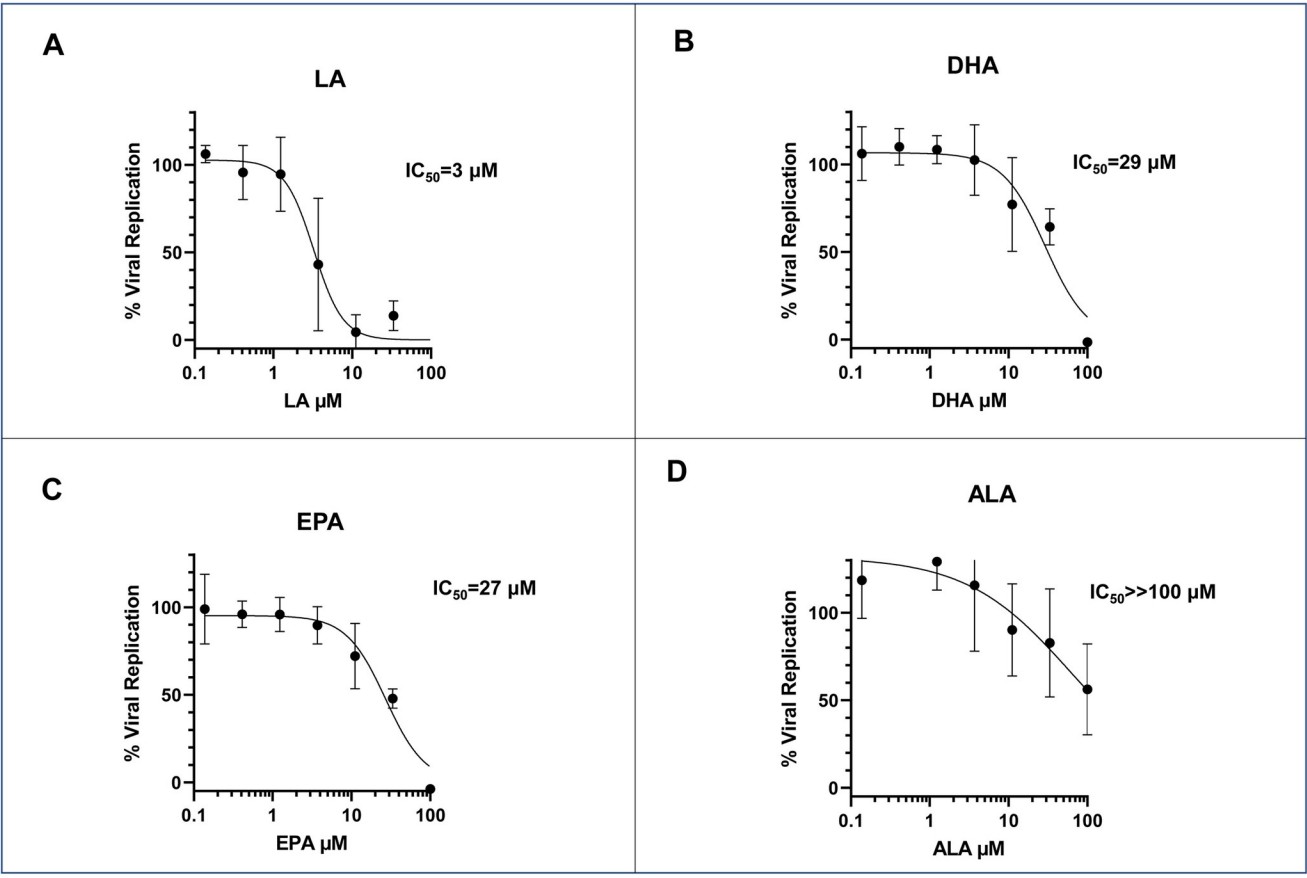

**Fig 3. Effect of different polyunsaturated fatty acids (PUFAs) in the inhibition of HCoV-229Eviral replication in MRC5 cells using antibody-based assay demonstrates significant inhibition of viral replication.** (A): Determination of the IC50value of linoleic acid (LA), demonstrated significant inhibition of viral replication. (B): Determination of the IC50value of docosahexaenoic acid(DHA), demonstrated significant inhibition of viral replication. (C): Determination of the IC50value of eicosapentaenoicacid(EPA), demonstrated significant inhibition of viral replication. (D): Determination of the IC50value of alpha-linolenic acid(ALA), demonstrated least inhibition.

also demonstrated significant inhibition of viral replication, while ALA demonstrated the least impact. As our initial intended use of PUFAs was a combination of VPA and DHA, we tested the combination of VPA with both DHA and LA to determine if the combination was mutually antagonist, additive, or supra-additive (e.g., synergistic). MRC5 cells were pre-incubated with VPA with and without LA or DHA at a fixed concentration of 25 µM. While 25 µM LA inhibited viral replication (Fig 4A), as expected, it did not significantly improve the $IC_{50}$ of VPA in the inhibition of viral replication, as the 95% confidence limits were grossly overlapping. In contrast, the combination of 25 µM DHA + VPA resulted in a substantial shift in the VPA $IC_{50}$, with nearly a 10-fold improvement in the inhibition of viral replication (Fig 4B), with a two-sided ANOVA of $p<0.001$ and completely non-overlapping 95% confidence limits for the $IC_{50}$. Please note that while the $IC_{50}$ of VPA was higher in the LA combination experiment than the DHA combination when the two VPA-only curves were compared across the two experiments (two-tailed nested t-test), they were not statistically different ($p = 0.41$). The differences were consistent with our known interassay variability in the measurements of viral replication.

We next turned our attention to whether the VPA+DHA combination was unique to VPA or represented a broader phenomenon present with other HDAC inhibitors. Depsipeptide and trichostatin were assessed for the impact of the combination with and without 25 μM DHA on viral replication. Fig 4C shows that Trichostatin A alone did not significantly inhibit viral replication, and the combination with DHA causes inhibition across the board, as expected. However, the inhibition of replication was only augmented at the very highest dose of Trichostatin A tested. In contrast, Depsipeptide demonstrated significant inhibition of viral replication; however, DHA only shifted the $IC_{50}$ from 0.8 to 0.1 nM and was not significant by two-tailed ANOVA ($p = 0.58$). While Trichostatin and Depsipeptide inhibit HDAC1 and HDAC2, trichostatin treatment of multiple cell lines has demonstrated a distinctly different gene expression pattern than depsipeptide [35]. This differential activity may explain why VPA and Depsipeptide have better antiviral activity as a monotherapy. Trichostatin's inhibition of HDACs is a direct effect through chelation of zinc in the active site, preventing histone

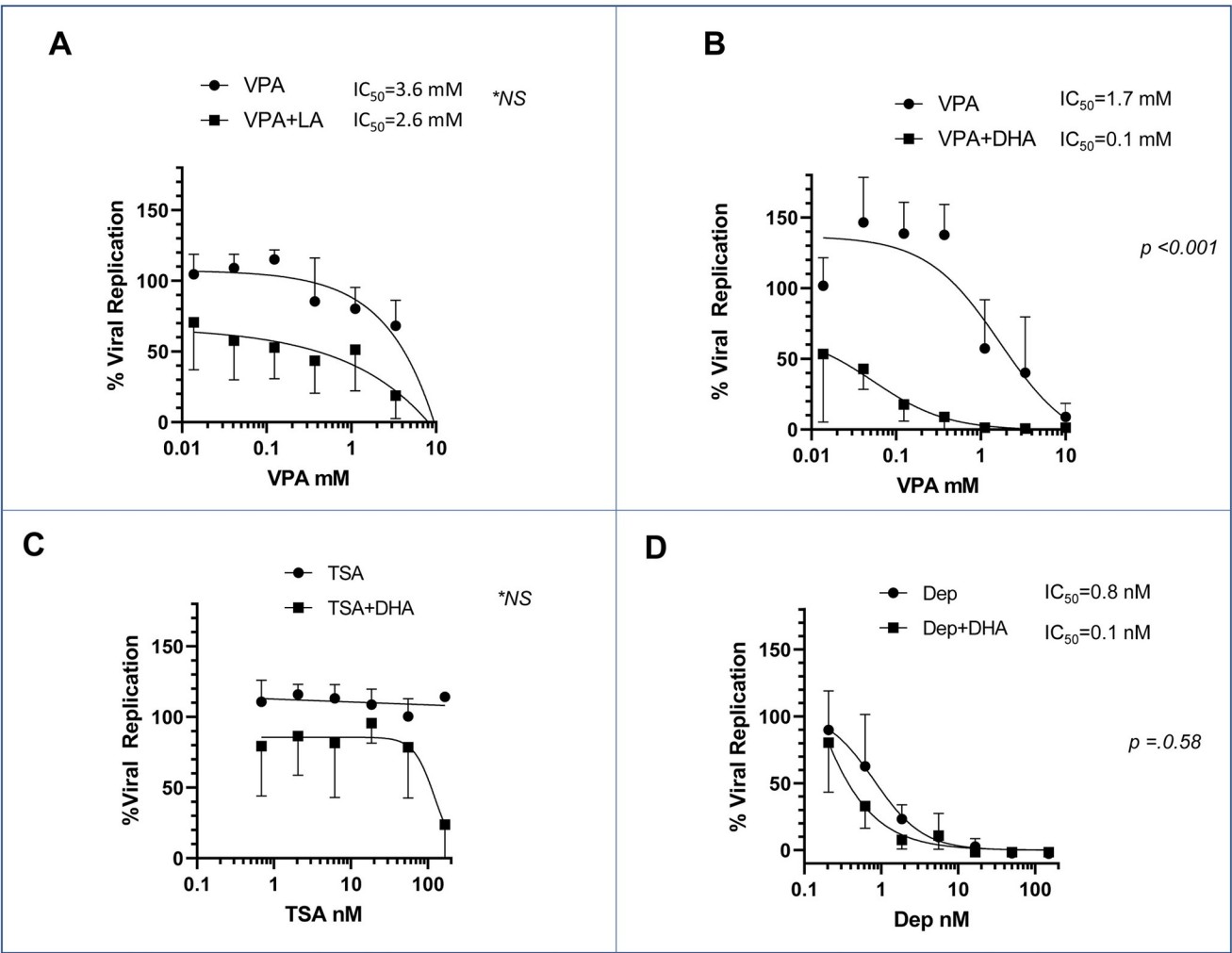

**Fig 4. Viral inhibition assay in combination of VPA with or without fixed dose (25 μM) of both DHA and LA.** (A): Determination of the IC50 value of VPA with or without linoleic acid (LA). (B): Determination of the IC50 value of VPA with or without docosahexaenoic acid (DHA). (C): Determination of the IC50 value of TrichostatinA(TSA) with or without docosahexaenoic acid (DHA). (D): Determination of the IC50 value of Depsipeptide (Dep) with or without docosahexaenoic acid (DHA).

unpacking, while VPA and Depsipeptide exert their inhibition more directly on the enzymatic activity of deacetylation, which potentially may have activity more broadly than just acetylated histones in the nucleus [36]. However, regardless of this differential activity in gene expression, the combination with DHA appears to confer a substantial synergy of antiviral activity.

## Gene expression of MRC5 cells is dramatically altered by HCoV-229E infection when treated with VPA+DHA

We tested whether DHA impacted VPA-regulated target gene expression or HCoV-229E viral replication. Twelve experiments were performed with the MRC5 cells, either in the presence or absence of drug (VPA or DHA or both) and virus (HCoV-229E). The virus was added at an MOI of 0.1 for one hour, and the drug (VPA, DHA, or both) was added after the viral incubation for 48 hours before RNA was harvested. In one experiment, MRC5 cells were "pre-incubated" with 0.5 mM VPA + 25 μM DHA for 24 hours before adding the HCoV-229E virus and then the drug combination continued for another 48 hours before RNA harvest. RNA from the cell lysates were analyzed for gene expression by RNAseq on an Illumina HiSeq 2000. Viral replication was determined by the amount of viral RNA detected expressed as the % of Total RNA from each experimental condition. As seen in Fig 5A, viral replication was not significantly impacted when VPA was added during viral incubation. In contrast, 25 μM DHA added immediately after viral incubation resulted in more than a 1/3 drop in viral RNA, consistent with our viral replication experiments. The contemporaneous addition of 0.1 mM VPA and 0.5 mM VPA with 25 μM DHA resulted in a further reduction of viral RNA, but the most pronounced effect was the combination of DHA and VPA with 24-hour preincubation.

Differential gene expression (DGE) was performed using iDEP 0.96 [14] to identify the top 2000 genes in each treatment group when compared to either untreated MRC5 cells, or MRC5 cells treated with virus without drug. In the case of DRUG only group, there were only 1914 genes differentially expressed, which was reduced further to between 500–650 differentially expressed genes when screened by a p-value of $< 0.05$ (Fig 5B) in the drug-treated group. A substantial differential gene expression pattern was observed due to viral infection in the second group treated with HcoV-229E (Fig 5B). K-means clustering was performed to identify six major gene clusters, and a scatterplot was made comparing the $Log_2$(fold change) expression for each treatment group versus their control group, as shown in Fig 5C and 5D. Notable in Fig 5B and 5D, the preincubation group of 0.5 mM VPA + 25 μM DHA treated for 24 hours prior to viral infection, resulted in the most extreme DGE. The Pearson correlation coefficient in Fig 5D starts at near 0 for the control MRC5 cells versus virus-infected cells for the 6 clusters and continues to be slightly positive in VPA and DHA-treated cells; however, when the combination of VPA and DHA is used, the slope is distinctly negative with the most profound change in the pre-incubated experiment, where the Pearson correlation coefficient is -0.7 (S2 Table). Visual confirmation of these substantial gene expression changes is noticeable in the color-coded clusters where Cluster B, C, and F significantly increase their gene expression in the pre-incubated drug combination relative to control. In contrast, Cluster A and E have fewer relative gene expression changes. These findings indicate that hundreds of genes are impacted in their response to HCoV-229E infection when pre-incubated with VPA and DHA in a manner that does not occur with either drug alone.

To better understand the overall impact of the various drug combinations on MRC5 gene expression, we performed Principle Component Analysis for the Differential Gene Expression to achieve a two-dimensional plot, representing the totality of the gene expression of each condition (e.g., dimensionality reduction) as shown in Fig 5E. In the drug-treated-only group, treatment with either 25 μM DHA or 0.1 mM VPA resulted in two relatively independent gene

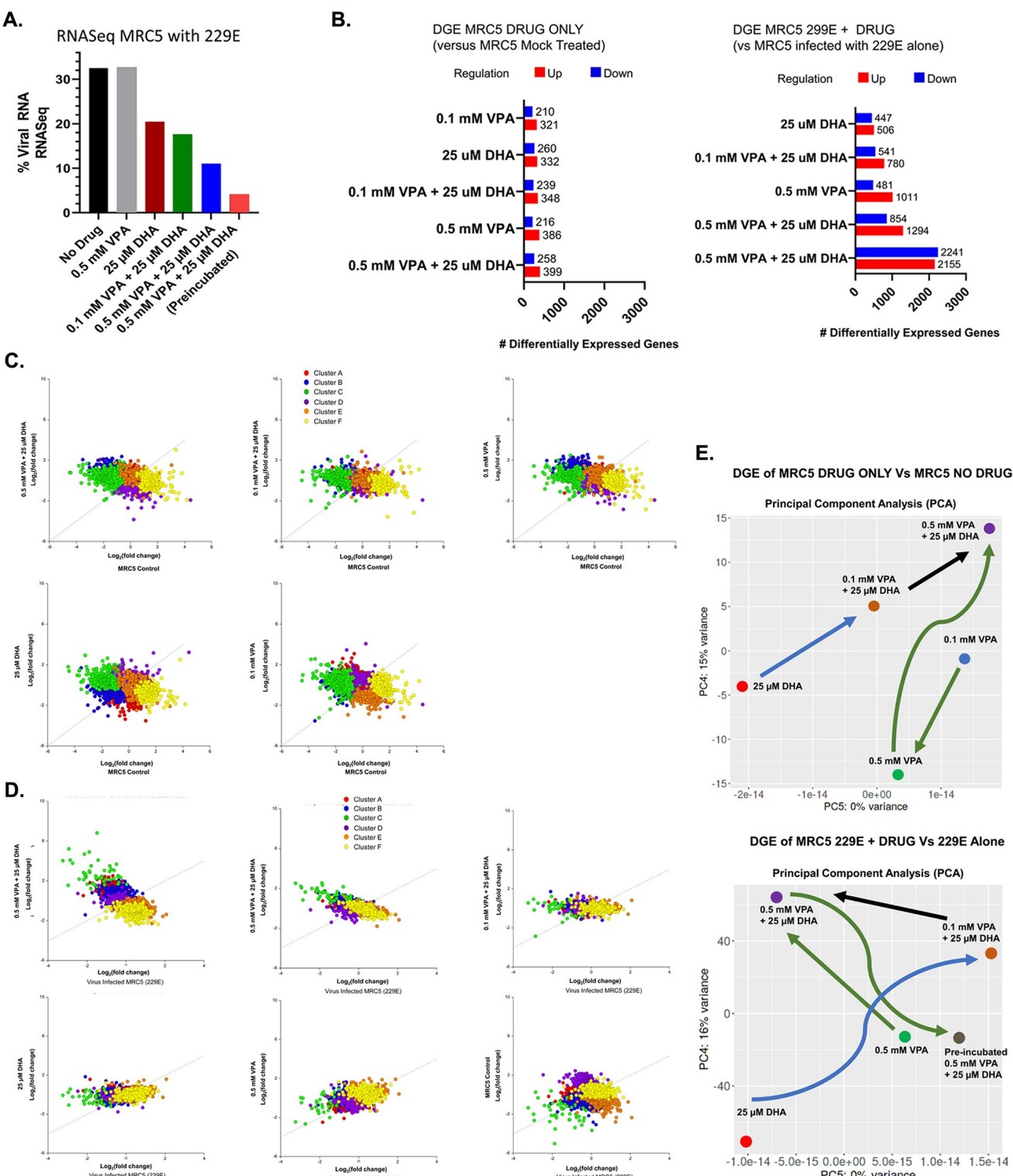

**Fig 5.** (A) Percentage of viral RNA sequence detected in RNA-seq from total RNA content of MRC5 cells infected with HCoV-229E at different experimental conditions and different drug combinations. (B) Number of differentially expressed genes (DEGs) identified in MRC5 cells when treated with different drug combinations. MRC5 cells, drug-only compared to mock-treated (Left). MRC5 cells infected with HCoV-229E treated with drug compared to MRC5 cells infected with HCoV-229E alone (Right). (C) K-means clustering scatterplot comparing the Log2(fold change) expression for each treatment group (VPA, DHA, VPA+DHA) versus the MRC5 control cells. (D) K-means clustering scatterplot comparing the Log2(fold change) expression for each treatment group (VPA, DHA, VPA+DHA) versus HCoV-229Einfected MRC5 cells. (E) Principle Component Analysis (PCA) of the differential gene expression obtained underdifferent treatment conditions.

expression profiles which were further impacted by the addition of either more VPA or the addition of VPA to the DHA, culminating with the most significant difference found by the combination of 0.5 mM VPA + 25 µM DHA. Similarly, when the virus is added, both the DHA and VPA-only patterns are distinctly different, converging with the combination and the pre-incubated combination providing the most significantly different condition. These findings indicate that treatment with DHA is a modifier of gene expression, not just binding to the virus in a hydrophobic pocket to affect uptake. Also, these results indicate that when virus infection occurs, the gene expression patterns of DHA and VPA are independent, and the pre-incubated combination is also substantially different than either alone-suggesting a form of synergy.

Finally, we sought to identify which molecular pathways are most impacted by the treatment with DHA and VPA. We utilized the Qiagen Ingenuity Pathway Analysis (IPA) to achieve this. The RNASeq data on MRC5 cells were entered into IPA and subjected to pathway analysis, with output including pathway, relative Z-score, and p-value. Data were plotted on a volcano plot, as shown in Fig 6A–6F, with each condition labeled, pairing each drug condition with drug along versus drug + virus. In the case of 25 µM DHA, viral replication induced positive Z-scores for Oxidative Phosphorylation, Glycolysis and NRF-2 Stress Response, and a negative Z-score for Micro- pinocytosis Signaling. VPA treatment alone (Fig 6C and 6D) induced pathways commonly associated with HDACi in general, but inflammatory pathways became dominant with the addition of the virus. The combination of the virus with 25 µM DHA + 0.5 mM VPA demonstrated a further induction of oxidative phosphorylation and enhanced energy metabolism but also showed a substantial down-regulation of Sirtuin Signaling

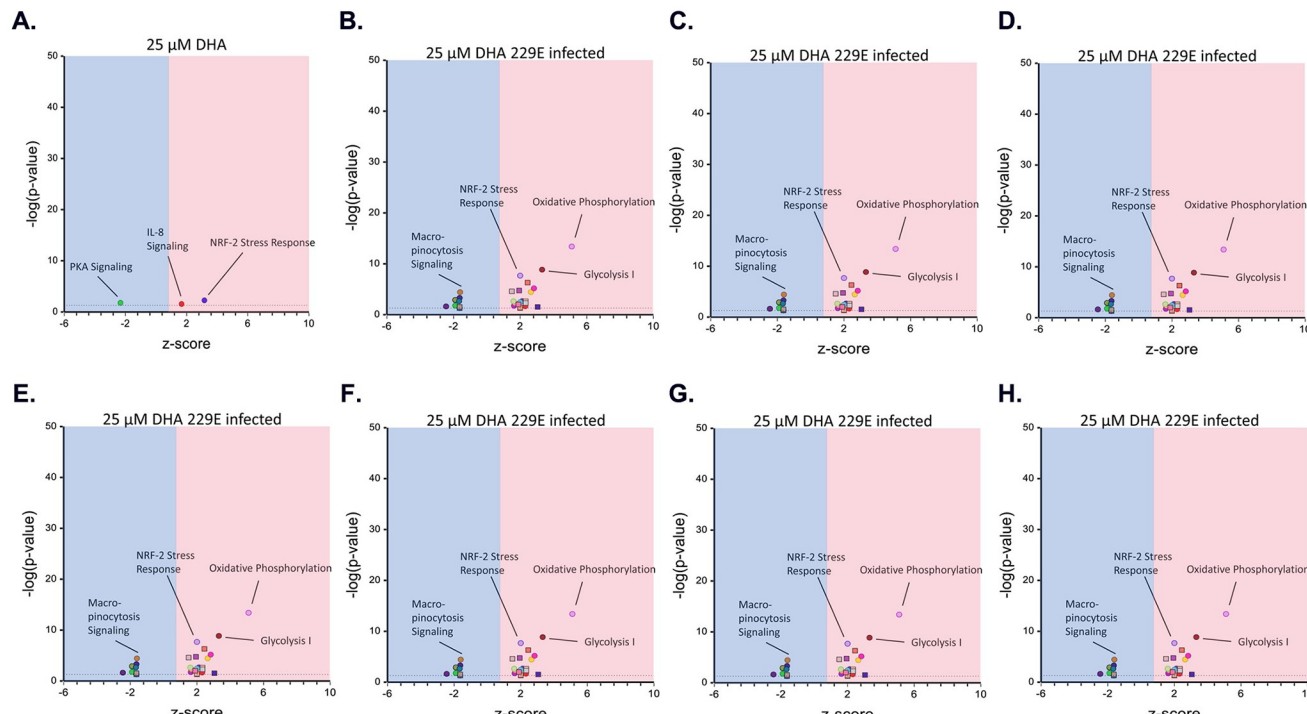

**Fig 6. Scatter plots of the molecular pathways obtained from ingenuity pathway analysis (IPA) under different treatment combinations of DHA and VPA.** MRC5 cells were treated with (A) 25 µM DHA, (B) 25 µM DHA and infected with HCoV-229E, (C) 0.5 mM VPA, (D) 0.5 mM VPA and infected with HCoV-229E, (E) 25 µM DHA + 0.5 mM VPA, (F) 25 µM DHA + 0.5 mM VPA and infected with HCoV-229E, (G) 25 µM DHA + 0.5 mM VPA and preincubated with HCoV-229E, (H) MRC5 Infected with HCoV-229E (no drug).

(Fig 6F), which is continued with preincubation (Fig 6G). The genes associated with each pathway are shown in S3 Table. These gene expression changes, in essence, demonstrate a previously unknown strong antiviral response induced by the combination of an HDACi and DHA. This antiviral effect is maintained even with doses of VPA which otherwise would not protect in the absence of DHA.

## VPA + DHA treatment of SARS-CoV2 inhibits viral replication

We tested whether the antiviral activity of DHA+VPA was present in SARS-CoV2. MRC5 cells were treated with SARS-CoV2 at an MOI of 0.1, and treated with VPA, DHA, DHA+VPA or control MRC5 without drug. SARS-CoV2 RNA was then detected by qRT-PCR (Fig 7A). Inhibition of viral replication was assessed by plotting the %Viral RNA detected compared to the control MRC5 cells without drug. As can be seen in Fig 7A, VPA, DHA and the combination of VPA+DHA all have substantial inhibitory activity against SARS-CoV2 at the doses tested (treated vs nontreated, p<0.001), with preincubation by 24 hours dropping the viral inhibitory effect from 20% to less than 10% though this did not reach statistical significance because of the low n (i.e., more than 50% reduction, $p = 0.145$, n = 3, two tailed t-test). Additional preincubation up to 5 days did not provide any additional effect. DHA conferred an inhibitory effect by itself, as it did with HCoV-229E, but preincubation did not improve the effect. Finally, the combination of VPA+DHA while highly effect at d0, was not more effective with preincubation though the experiment had sufficiently wide error bars that such an effect might have been missed at these doses. MRC5 does NOT have the canonical receptor ACE2; hence, viral binding, uptake and spreading are markedly diminished in MRC5 cells. We estimated that less than 1% of the total isolated RNA was SARS-CoV2, consistent with the MOI, viral replication in the cells which permitted viral uptake, but without viral spreading due to the lack of receptor. Nonetheless, there was sufficient uptake and replication to result in a roughly 5000 fold increase of viral RNA over input RNA from the virus added to the culture. RNASeq was conducted on the cell lysates as a secondary confirmation of inhibition of viral replication as demonstrated in Fig 7B. These data demonstrate that VPA, DHA and VPA+DHA all have

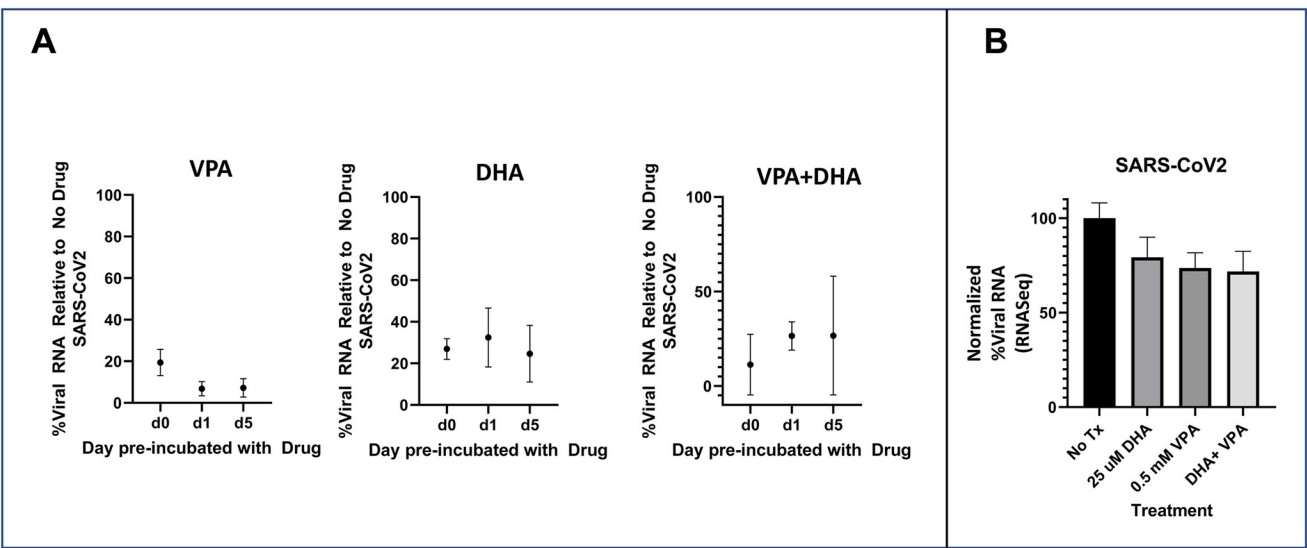

**Fig 7.** (A.) Percentage of viral RNA detected by qRT-PCR in MRC5 infected with SARS-CoV2 when treated with VPA, DHA, and VPA+DHA. (B.) Percentage of normalized viral RNA detected by RNASeqfrom the cell lysates treated with VPA, DHA and VPA+DHA.

inhibition of viral replication. However, the degree of inhibition may be underestimated due to the small amount of viral RNA detectable. These data demonstrate nonetheless, that despite the lack of ACE2 (which is known to be downregulated by VPA), VPA as well as the combination of VPA+DHA inhibit viral replication and provide the foundation for a novel antiviral strategy against coronaviruses.

## Discussion

The novel coronavirus (SARS-CoV2) pandemic is one of the most impactful infections in modern history. While a historic number of people have been infected and died from this disease, the worldwide response to vaccine development has been equally impressive. Unfortunately, incomplete vaccination and persistent viral mutations have hampered attempts to eradicate this disease, and ultimately vaccination therapy alone is not a viable long-term strategy in a world wide pandemic. Broad-spectrum antiviral drug development must be pursued and optimized to combat this virus, its variants, and other similar highly transmissible respiratory viruses to alter the spread of these agents. Early bioinformatic methodologies greatly facilitated strategies for antiviral drug development, which demonstrated the critical pathways of interaction between SARS-CoV2 proteins and cellular proteins [10, 11, 13, 20, 37–46]. Among the potential early targets identified by this methodology was HDAC2 [47], found to interact with nsp5 [10]. Efforts have been underway to evaluate the potential efficacy of HDAC inhibitors on Coronavirus replication [21, 48]. Among the drugs studied was the short-chain fatty acid Valproic acid [2], a commonly used antiseizure agent also used for bipolar disorder [49, 50] and migraine headaches. VPA was first approved in the 1970s and has a well-established safety profile and therapeutic index. While there are well-known potential toxicities, including fetal teratogenicity, somnolence, thrombocytopenia, and rare episodes of hepatitis and pancreatitis, it has been well tolerated by most patients over prolonged periods.

The most significant concern with VPA is teratogenicity which can cause an increase in the risk of neural tube defects in the fetuses of pregnant women [51]. This activity may be related to the HDAC2 inhibitory activity of VPA [52]. Early *in vitro* testing of Valproic acid against SARS-CoV2 [10] indicated no measurable inhibition of viral infection. In retrospect, however, these studies were performed in a dose range of VPA (μM range) significantly less than the $IC_{50}$ of HDAC2 inhibition [52] (mM range). *In vitro* studies of VPA in the mM range have demonstrated that treatment of cells reduces expression of the SARS-CoV2 receptor ACE2 as well as IL-6 and ICAM- 1 [53], although the high dose requirements in high-throughput viral replication assays hindered enthusiasm for potential clinical use. High throughput screening assays, however, are developed to identify agents which inhibit SARS-CoV2 replication *__directly__*. HDAC inhibition is an *__indirect__* strategy. These agents must first inhibit histone deacetylation to alter gene expression in susceptible gene loci. Induction of altered gene expression takes at least 24 hours [52], as demonstrated in Fig 2. In many SARS-CoV2 assays, viral replication is completed within 24 hours.

Some small studies have recently suggested a potential benefit of VPA against COVID-19, despite dose ranges well below the predicted $IC_{50}$. For instance, Moreno-Pérez *et al.* demonstrated that of the 691 patients taking VPA in their case-control study, only 12 (1.7%) tested positive for SARS- CoV2, while the controls tested positive at a rate of 2.2%, and only one of the twelve VPA patients required hospitalization (0.14%) while the control hospitalization rate was 0.26% [54]. Serum VPA levels were available for 442 of the 691 patients during the study period, and when corrected for the seizure disorder therapeutic range, the calculated OR for contracting COVID-19 was 0.218 in the therapeutic VPA cohort. While these dramatic reductions were statistically significant, the clinical relevance of these findings was

lessened by the small number of patients in the study. Similarly, Collazos *et al.* found in a retrospective study of 165 VPA-treated patients admitted to the hospital for COVID-19 that the VPA-treated patients had a shorter duration of symptoms, lower in-hospital respiratory worsening, lower ICU admissions, and fewer pulmonary infiltrates than comparable controls matched for sex, age, and date of admission [55]. The mortality, however, was not statistically different. Like before, the small number of patients made interpretation difficult. Here we report the only large cohort study of the association of VPA with COVID-19 and find a significant association of VPA use with diminished disease contraction and severity. Using a national database from Optum, we discovered that patients prescribed VPA have a 25% decreased risk of contracting COVID-19 in an exact-matched multivariate analysis determined by nucleic acid testing. Since the dose required for antiviral activity is relatively high (>85 μg/mL, or 0.6 mM) in our *in vitro* testing, such a protective effect is impressive as fewer than 25% of VPA-treated patients are in the predicted optimal therapeutic range for antiviral activity.

Once we discovered a putative protective effect, we then sought to understand why this activity, as initially predicted, was not demonstrated in prior *in vitro* viral replication assays. As we initially suspected, the $IC_{50}$ for VPA in a conventional coronavirus replication assay (HCoV-229E was used as a prototype), was nearly 5 mM, which translates to over 720 μg/mL-nearly six times above the toxicity level in humans (125 μg/mL). The dose required in these assays was considered toxic, and no further evaluation would have been recommended. However, HDAC inhibitors work primarily by altering transcription in a process that often takes >24 hours. We reasoned that the protective effect of VPA in preventing conversion to COVID-19 positive may have been a manifestation of chronicity of VPA administration, in which gene induction occurs at a lower dose of VPA as previously identified in cancer therapeutic models [6]. We performed time course experiments in coronavirus permissive cell lines and demonstrated that gene expression profile changes require at least 24 hours before they can be detected, and in some cases, they are maximal between 72–96 hours (Fig 2). A comparison of the SARS-CoV2 interacting genes demonstrates that the expression of multiple genes was reduced, involving at least four SARS-CoV2 pathways (Fig 2B & 2C). Incubation of the HCoV-229E permissive cell line MRC5 with Valproic acid for at least 24 hours prior to infection demonstrated that the $IC_{50}$ for Inhibition of viral replication was reduced by nearly 10-fold, down to roughly 0.6 mM (corresponding to 85 μg/mL serum levels). Significantly, *in vitro* drug concentrations do not necessarily correlate precisely with serum drug levels, as serum drug delivery is a far more complex process with drug partitioning into different functional compartments and dynamically changing equilibrium of free drug based on the tightly regulated carrier proteins. Nonetheless, our i*n vitro* results are consistent with the 25% reduction in COVID- 19 positive testing among patients on active Valproic acid treatment, as less than 25% of patients have VPA levels more than 85 μg/mL. This alone would be sufficient cause for further study; however, we sought to see if the activity of VPA could be enhanced by other methods to lessen potential clinical toxicity.

Recent studies have suggested that patients who consume dietary supplements, including omega- 3 fatty acids (FAs), have a less severe COVID-19 disease course [56–58]. An intervention trial with omega-3 FAs infused intravenously demonstrated a slight improvement in the disease course in critically ill patients, though the study was small and did not reach statistical significance [59]. Since the polyunsaturated fatty acid PUFA DHA has been demonstrated to decrease the side effects of Valproic acid, we assessed the combination *in vitro* to determine if there was any alteration of the antiviral activity of VPA. Surprisingly, the combination enhanced the antiviral activity, reducing the dose requirement of VPA to a level of 0.1 mM, a dose easily achievable in human patients without the side effects most commonly associated

with the higher doses required for antiseizure activity (Fig 4B). DHA does not have any HDAC inhibitory activity but does have antiviral activity.

The combination potently inhibits coronavirus replication both with HCoV-229E and SARS-CoV2. The combination of DHA with other HDAC inhibitors demonstrates that enhanced activity is maintained with other HDAC2 inhibitors. Additionally, the gene expression profiles of MRC5 cells treated with VPA, DHA, or VPA + DHA both in the presence or absence of coronavirus HCoV-229E demonstrate that the additional antiviral activity is not a consequence of further HDAC inhibition. Instead, it appears that the combination activates pathways involved in interferon and downstream related genes with a scatterplot demonstrating a profound change in the overall gene expression profile to inhibit viral replication. Hence, this combination represents a novel target for Coronavirus antiviral therapy—and potentially an entirely new class of antiviral combination therapies.

Linoleic acid (LA) is an omega-6 polyunsaturated fatty acid (PUFA). It is also considered an essential fatty acid as it can give rise to multiple other fatty acid metabolites, including the omega- 3 fatty acids- alpha-linoleic acid (ALA), eicosapentaenoic acid (EPA), and docosahexaenoic acid (DHA). Toelzer et al. demonstrated that the receptor binding protein (S, or Spike) of SARS-CoV2 has a hydrophobic pocket capable of sequestering linoleic acid (LA), causing a conformational change impacting viral infectivity [49]. Combining LA with Remdesivir resulted in a significant improvement in the inhibition of coronavirus replication. The combination of LA with VPA did not have a similar effect in HCoV-229E; however, sequence comparison of the binding region between SARS-CoV2 and HCoV-229E predicted a potential lack of LA binding in HCoV-229E. Furthermore, MRC5 lacks the ACE2 receptor [60], preventing testing with SARS-CoV2 in MRC5 cells.

Additionally, HCoV-229E uses the human aminopeptidase N as its receptor [61], so the LA lipid binding region identified by Toelzer *et al.* may not be a universally applicable strategy for all coronaviruses. Combining VPA with all three primary omega-3 fatty acids (ALA, DHA, EPA) demonstrates that the selectivity of the combination favors DHA. While LA may help inhibit SARS- CoV2 by directly affecting viral conformation, DHA causes gene expression changes, providing a broader strategy as it is not as susceptible to COVID-19 variants such as D614G, lacking the susceptibility to LA binding [31].

In summary, epidemiological data indicate that patients who take VPA have a decreased rate of COVID-19 test positivity, a decreased rate of ER visits, a decreased rate of hospitalization if they contract COVID-19, a decreased rate of ICU admissions, and a decreased rate of mechanical ventilation. Given the time window during which this data was extracted, this effect continued through the transition from the alpha to the delta variant-indicating an overall effect not lost with virulence mutations. Serum VPA testing nationally indicates that only about 25% of the patients who take VPA would have had sufficient serum levels for Inhibition of SARS-COV2, according to the *in vitro* data. However, caution must be taken with such data, as these are associations do not show causality particularly when the in vitro prediction of drug levels is driven by non-applicable models of infection. The small branched-chain fatty acid VPA combined with the omega-3 PUFA DHA results in a marked antiviral activity against coronaviruses. This treatment appears effective due to activating a large set of antiviral genes involving the re-activation of innate immune genes, including many interferon-regulated genes. This strategy is attractive as it may retain efficacy even with the emergence of mutant coronavirus variants while minimizing disease severity and spread. These studies lay the groundwork for further clinical evaluation of VPA + DHA for treating coronaviruses in general and provide the additional foundation for combination therapeutics of PUFAs and HDACi.

## Supporting information

**S1 Fig. Determination of IC$_{50}$ and maximum induction of VPA using Vero cells.** (A.) IC$_{50}$ curve of VPA in directly inhibiting HDAC2 activity in Vero cells. (B.) Top -Protein expression level of p21 upon treatment of VPA in different time points. Bottom- Time course induction level of p21 by 2.55mM of VPA in Vero cells.
(TIF)

**S1 Table. Distribution of Serum VPA levels during the second quarter of 2021 as collected by LabCorp, Inc.**
(PPTX)

**S2 Table. Pearson correlation coefficient values and 95% confidence interval for the drug-only group (top) and virus-infected + drug (bottom) under the various tested treatment conditions.**
(PPTX)

**S3 Table. Table of different genes associated with the respective pathways in IPA.**
(PPTX)

**S1 Raw images.**
(PDF)

## Acknowledgments

Lab Corp, Inc provided data on serum Valproic acid levels in patients nationally. Consultation with the Trial Innovation network assited in development of clinically pragmatic aspects of this report (NCATS Grants U24TR001609 and U24TR004440.

## Author Contributions

**Conceptualization:** Ronald Rodriguez.

**Data curation:** Pankil Shah.

**Formal analysis:** Amanda Watson, Pankil Shah, Geeta Joshi, Wasim H. Chowdhury, Dean Bacich, Daniel Hanley.

**Funding acquisition:** Ronald Rodriguez.

**Investigation:** Amanda Watson, Doug Lee, Sitai Liang, Dean Bacich, Peter Dube, Yan Xiang, Luis Martinez-Sobrido.

**Supervision:** Ronald Rodriguez.

**Writing – original draft:** Ronald Rodriguez.

**Writing – review & editing:** Geeta Joshi, Ediri Metitiri, Ronald Rodriguez.

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
