## [Decision Letter · Decision Letter 0]

8 May 2024

PONE-D-24-10375Valproic acid Use is Associated with Diminished Risk of Contracting COVID-19, and Diminished Disease Severity: Epidemiologic and in vitro Analysis Reveal Mechanistic InsightsPLOS ONE

Dear Dr. Rodriguez,

Thank you for submitting your manuscript to PLOS ONE. This study is  a valuable contribution to the field. After careful consideration, we feel that it has merit but needs minor modifications. Therefore, we invite you to submit a revised version of the manuscript that addresses the points raised during the review process specifically the comments from the Reviewer #1.

We look forward to receiving your revised manuscript.

Kind regards,

Mrinmoy Sanyal, PhD

Academic Editor

PLOS ONE

Journal Requirements:

"This work was funded by the Abraham and Linda Littenberg Foundation, the DHR Endowment and the UT Health San Antonio Dean’s Fund"

4. We note you have included a table to which you do not refer in the text of your manuscript. Please ensure that you refer to Table 4 in your text; if accepted, production will need this reference to link the reader to the Table.

5. We notice that your supplementary figures "Figure S1" are uploaded with the file type 'Figure'. Please amend the file type to 'Supporting Information'. Please ensure that each Supporting Information file has a legend listed in the manuscript after the references list."

**Additional Editor Comments:**

Thank you for the valuable contribution. Please address the comments from the Reviewer# 1.

Reviewers' comments:

Reviewer's Responses to Questions

**Comments to the Author**

1. Is the manuscript technically sound, and do the data support the conclusions?

Reviewer #1: Yes

Reviewer #2: Yes

2. Has the statistical analysis been performed appropriately and rigorously? 

Reviewer #1: Yes

Reviewer #2: Yes

3. Have the authors made all data underlying the findings in their manuscript fully available?

Reviewer #1: Yes

Reviewer #2: Yes

4. Is the manuscript presented in an intelligible fashion and written in standard English?

Reviewer #1: Yes

Reviewer #2: Yes

5. Review Comments to the Author

Reviewer #1: In the manuscript titled “Valproic acid Use is Associated with Diminished Risk of1 Contracting COVID-19, and Diminished Disease Severity:2 Epidemiologic and in vitro Analysis Reveal Mechanistic Insights”, the authors attempted to provide understandings to use VPA as an antiviral agent against coronaviruses. The study is interesting and opens up the possibility of using VPA in combination with DHA as a potential drug to treat COVID. However, I have a few minor concerns.

Figure S1B should be cited in the manuscript after the sentence (Line 444, page 24)

Conclusion sentence for Figure 2 (Lines 476-478, page 25) may be rephrased for better understanding to the audience.

I feel authors should provide higher resolution image for figures 2, 5C,5D and 6

Reviewer #2: Watson e et al in a very interesting manuscript confirmed the suspected protective effect of valproic acid (VPA) on SARS-Cov-2 infection .Firstly they confirmed this protective effect using real life data taken from a large American cohort of patients on chronic VPA therapy. They observed a 25% decreasing risk of contracting COVID-19 or developing serious COVID-19 -associated clinical complications (significantly less ER visits, Inpatient admissions, mechanical ventilation needs, ICU admissions in those exposed to VPA).Surprisingly the authors observed that only 25% of VPA-treated patients had VPA plasma levels within the therapeutical range estimated for VPA for antiviral activity. They explained this contradiction based on the fact that the protective effect of VPA on SARS-CoV-2 replication, due to genes induction, might be higher in patients exposed chronically to VPA. Furthermore the authors confirmed that the antiviral effect of VPA requires from 24h -96 h of exposition in an elegant in vitro experiment using HCov-229E virus .

The antiviral effect of omega-3 fatty acids on COVID-19 was also explored. The polyunsaturated fatty acid (PUFA) DHA showed the highest antiviral effect among the different linoleic acid derivates in in vitro assays against the same HCov-229E virus. DHA causes gene expression changes, a complementary mechanism to that played by VPA. Then they showed a synergistic effect of the combination of VPA+ DHA on HCov-229E and SARS-CoV-2 viruses replication. This antiviral effect synergistic of VPA+ DHA is due to the reactivation of innate immune genes, including interferon-regulated genes.

This excellent works paves the way for further clinical evaluation of the combination of VPA + DHA to treat COVID-19. Protection of vaccines against SARS-CoV-2 is time-limited requiring frequent boosters and does not provide a completed cover. This attractive combination of VPA-DHA to be used against coronaviruses is worth to be tested already in phase I studies although with great care considering the well-known toxicities of VPA including teratogenicity

6. PLOS authors have the option to publish the peer review history of their article (what does this mean?). If published, this will include your full peer review and any attached files.

Reviewer #1: No

Reviewer #2: No

---

## [Author Response · Author response to Decision Letter 0]

18 Jun 2024

To the Editor:

Thank you for recognizing our work and giving us the opportunity to address the issues raised. We

have addressed each point raised during the review process.

Editor Item #1:

We note you have included a table to which you do not refer in the text of your manuscript. Please

ensure that you refer to Table 4 in your text; if accepted, production will need this reference to link

the reader to the Table.

Response: Thank you for pointing it out. Now we have referenced the Table 4 in the main text

(Line 424 , Page 22).

Editor Item #2:

We notice that your supplementary figures "Figure S1" are uploaded with the file type 'Figure'.

Please amend the file type to 'Supporting Information'. Please ensure that each Supporting

Information file has a legend listed in the manuscript after the references list."

Response: We have now uploaded Figure S1 to file type to 'Supporting Information'. Legend is

listed in the manuscript after the reference list.

Item #1 Reviewer #1: In the manuscript titled “Valproic acid Use is Associated with Diminished

Risk of1 Contracting COVID-19, and Diminished Disease Severity:2 Epidemiologic and in vitro

Analysis Reveal Mechanistic Insights”, the authors attempted to provide understandings to use VPA

as an antiviral agent against coronaviruses. The study is interesting and opens up the possibility of

using VPA in combination with DHA as a potential drug to treat COVID. However, I have a few minor

concerns.

We thank reviewer1 for recognizing the potential of our research and encouraging comments. We

have provided point-to-point response.

Figure S1B should be cited in the manuscript after the sentence (Line 444, page 24)

• Response: Thank you for pointing it out. Now, Figure S1B is cited as suggested.

Conclusion sentence for Figure 2 (Lines 476-478, page 25) may be rephrased for better

understanding to the audience.

• Response: Thank you for the comment. The sentence is rephrased for the clarity.

I feel authors should provide higher resolution image for figures 2, 5C,5D and 6

• Response: We have provided higher resolution images for the figures.

Reviewer #2:

We thank Reviewer 2 for encouraging comments.

---

## [Decision Letter · Decision Letter 1]

2 Jul 2024

Valproic acid Use is Associated with Diminished Risk of Contracting COVID-19, and Diminished Disease Severity: Epidemiologic and in vitro Analysis Reveal Mechanistic Insights

PONE-D-24-10375R1

Dear Dr. Rodriguez,

We’re pleased to inform you that your manuscript has been judged scientifically suitable for publication and will be formally accepted for publication once it meets all outstanding technical requirements.

Kind regards,

Mrinmoy Sanyal, PhD

Academic Editor

PLOS ONE

Reviewers' comments:

Reviewer's Responses to Questions

**Comments to the Author**

1. If the authors have adequately addressed your comments raised in a previous round of review and you feel that this manuscript is now acceptable for publication, you may indicate that here to bypass the “Comments to the Author” section, enter your conflict of interest statement in the “Confidential to Editor” section, and submit your "Accept" recommendation.

Reviewer #1: All comments have been addressed

2. Is the manuscript technically sound, and do the data support the conclusions?

Reviewer #1: Yes

3. Has the statistical analysis been performed appropriately and rigorously? 

Reviewer #1: N/A

4. Have the authors made all data underlying the findings in their manuscript fully available?

Reviewer #1: Yes

5. Is the manuscript presented in an intelligible fashion and written in standard English?

Reviewer #1: Yes

6. Review Comments to the Author

Reviewer #1: (No Response)

7. PLOS authors have the option to publish the peer review history of their article (what does this mean?). If published, this will include your full peer review and any attached files.

Reviewer #1: **Yes: **Aiswarya Sethumadhavan

---

## [Editor Report · Acceptance letter]

25 Jul 2024

PONE-D-24-10375R1 

PLOS ONE

Dear Dr. Rodriguez, 

I'm pleased to inform you that your manuscript has been deemed suitable for publication in PLOS ONE. Congratulations! Your manuscript is now being handed over to our production team.

Kind regards, 

on behalf of

Dr. Mrinmoy Sanyal 

Academic Editor

PLOS ONE